

# CLARA-A2: The second edition of the CM SAF cloud and radiation data record from 34 years of global AVHRR data

**Karl-Göran Karlsson[1], Kati Anttila[2], Jörg Trentmann[3], Martin Stengel[3], Jan Fokke Meirink[4], Abhay Devasthale[1], Timo Hanschmann[3], Steffen Kothe[3], Emmihenna Jääskeläinen[2], Joseph Sedlar[1], Nikos Benas[4], Gerd-Jan van Zadelhoff[4], Cornelia Schlundt[3], Diana Stein[3], Stephan Finkensieper[3], Nina Håkansson[1] and Rainer Hollmann[3]**

[1]{Swedish Meteorological and Hydrological Institute (SMHI), Norrköping, Sweden}

[2]{Finnish Meteorological Institute (FMI), Helsinki, Finland}

[3]{Deutscher Wetterdienst (DWD), Offenbach, Germany}

[4]{Royal Netherlands Meteorological Institute (KNMI), De Bilt, The Netherlands}

*Correspondence to*: Karl-Göran Karlsson (Karl-Goran.Karlsson@smhi.se)

**Abstract**

The second edition of the satellite-derived climate data record CLARA ("The CM SAF cLoud, Albedo and surface RAdiation dataset from AVHRR data" - second edition denoted CLARA-A2) is described. The data record covers the 34-year period from 1982 until 2015 and consists of cloud, surface albedo and surface radiation budget products derived from the AVHRR (Advanced Very High Resolution Radiometer) sensor carried by polar-orbiting, operational meteorological satellites. The data record is produced by the EUMETSAT Climate Monitoring Satellite Application Facility (CM SAF) project as part of the operational ground segment. Its upgraded content and methodology improvements since edition 1 are described in detail as well as some major validation results. Some of the main improvements of the data record come from a major effort in cleaning and homogenising the basic AVHRR level 1 radiance record and a systematic use of CALIPSO-CALIOP cloud information for development and validation purposes. Examples of applications studying decadal changes in Polar Summer surface albedo and cloud conditions, as well as global cloud redistribution patterns, are provided.

## 1 Introduction

Global distribution of cloudiness and existing cloud feedback on the radiative forcing continue to be important topics in climate research. Uncertainties in the description and understanding of both topics are considered to be



dominant in explaining the spread among climate models in their prediction of current and anticipated climate
change (Webb et al., 2013; Vial et al., 2013). In parallel, better knowledge and monitoring of global cloudiness
and radiation are also required for a successful increased utilisation of renewable energy sources, such as solar
power plants (Šúri et al., 2007). In order to address requests and challenges in these and adjacent fields by a
systematic utilization of satellite measurements, the Climate Monitoring Satellite Application Facility (CM SAF)
was formed in 1998 by the European Organisation for the Exploitation of Meteorological Satellites
EUMETSAT. The CM SAF project was introduced and described in detail in the ACP journal by Schulz et al.
8    (2009).

CM SAF (www.cmsaf.eu) aims at developing capabilities for a sustained generation and provision of Climate
Data Records (CDRs) derived from operational meteorological satellites. The ultimate aim is to make the
resulting data records suitable for the analysis of climate variability and the detection of climate trends.
Examples of important guidelines for the compilation of CDRs are, (1) to apply the highest standards and
guidelines as lined out by the Global Climate Observing System (GCOS), (2) to process satellite data within a
true international collaboration benefiting from developments at international level and, (3) to perform intensive
validation and improvement of the CM SAF CDRs, including a major role in data record assessments performed
by research organizations such as the World Climate Research Programme (WCRP).
One of CM SAF's CDRs is CLARA: "The CM SAF cLoud, Albedo and surface RAdiation dataset from
AVHRR data". It is based on data from the Advanced Very High Resolution Radiometer (AVHRR) operated
onboard the polar orbiting NOAA satellites as well as by the MetOp polar orbiters operated by EUMETSAT
since 2006. AVHRR offers one of the longest satellite observation records, with its first measurements starting
already in 1978. The first edition of CLARA (CLARA-A1) was released in 2012 and it was described in the
ACP journal by Karlsson et al. (2013). This paper describes improvements and other features of the second
edition, CLARA-A2, which was released in 2016. In addition to the description of the data record, an early
analysis of some of its components with particular relevance for climate studies is also provided with the purpose
of promoting more in depth follow-on studies in the near future.
The basic AVHRR radiance measurements were previously described in detail by Karlsson et al. (2013).
Consequently, Section 2 describes only the extension of the data series since CLARA-A1 and some further
modifications to improve calibration and homogenisation of the entire data record. Section 3 includes general
descriptions on how the data record was compiled and Sections 4-6 explain the most significant improvements
made to retrieval methods for the three different groups of parameters together with some validation results. For
the latter, some focus has been on extensive inter-comparisons being made to space-borne active lidar cloud
retrievals (CALIPSO-CALIOP) and to other existing satellite-based data records (e.g., PATMOS-x and
MODIS).  Section 7 discusses potential applications and provides some examples of analyses possible with a
continuous, homogeneous 34-yr data record. Finally, Section 8 summarises the main features of the data record
and presents future plans.



## 2    Extension and homogenisation of the historic AVHRR data record

The basic AVHRR radiance measurements (level 1 observations) used in CLARA-A2 are described in detail by Karlsson et al. (2013). However, the temporal coverage is now extended with six additional years (2010-2015) resulting in a total length of 34 years (1982-2015). Figure 1 illustrates all satellites and their respective measurements periods for the CLARA-A2 climate data record. From this figure it is clear that the observational coverage varies considerably; only one satellite in orbit providing measurements during the 1980s and early 90s, until the last decade where at least four simultaneous satellites were present (with a peak of 6 satellites available simultaneously in 2009). Additionally, orbital drift, leading to changing local observation times, contributes to create rather varying observation conditions during the period. However, some sub-setting of the data could still yield relatively homogeneous observation conditions. For example, if choosing exclusively afternoon satellites (which is possible with the CLARA-A2 data record) a quite homogeneous and stable time series of observations can be achieved.

The AVHRR instrument was initially built for operational global weather monitoring purposes, not for climate monitoring. This means that the radiometric accuracy and the stability of radiance measurements are sometimes problematic for some early satellites in the time series. In addition, NOAA archiving of data has its own problems with spurious occurrences of gaps, duplications and corrupt data. Consequently, a substantial effort in the preparation of CLARA-A2 has been made to correct and homogenize the entire radiance (level 1) record. A special pre-processing tool (PyGAC) was developed for these purposes, described in detail by Devasthale et al. (2016). Some of the most important aspects have been the following:

- Removal of corrupt data
- Data rescue of data with incorrect header definitions
- Removal of duplicated orbits
- Removal of overlap between orbits
- Homogenization of visible calibration by removal of trends and performing inter-calibrating between satellites (based on the method by Heidinger et al. (2010), but extended with more satellites and with MODIS Collection 6 as reference data)
- Improving accuracy of infrared calibration (compared to CLARA-A1) by using a more accurate treatment of calibration target data
- Applying median filters to AVHRR channel 3b (at 3.7 microns) brightness temperatures for reducing the impact of high noise levels for satellites NOAA-7 to NOAA-14
- Removal of partially corrupt orbits in periods with AVHRR scan motor problems (primarily between years 2001-2005; this was mostly based on manual inspection efforts since operational data flagging does not cover this problem sufficiently well)

The overall impact of these measures resulted in the exclusion of approximately 6 % of all original level 1 data in the NOAA archive from processing. The work with improving the AVHRR level 1 data record (or the Fundamental Climate Data Record – FCDR) has been performed within the framework of the WMO project SCOPE-CM (http://www.scope-cm.org/) and the ESA Cloud_cci project (http://www.esa-cloud_cci.org).





3    **Product overview highlighting changes in product aggregation since CLARA-A1**

The CLARA-A2 CDR is based on instantaneous AVHRR Global Area Coverage (GAC) retrievals (i.e., for every orbit at approximately 4 km horizontal swath resolution in nadir) which have been aggregated to derive the final spatio-temporally averaged data records. Since CLARA-A1, an important change for the cloud products is the introduction of globally resampled daily composites (level-2b) as the basis for computation of final level-3 products. The level-2b data representation (introduced by Heidinger et al., 2014) is motivated by the inhomogeneous global coverage of polar sun-synchronous satellite data. Each polar satellite offers around 14 evenly-distributed observations per day for each location near the pole while when passing the equator each location is observed only twice, approximately 12 hours apart. The idea with the introduction of the level-2b data representation is to form a more homogeneous data record having only two observations per day per satellite for each location globally. The alternative to use all observations for Level-3 products (as was done for CLARA-A1) results in a much skewed distribution of the observations because of the inhomogeneous observation frequency (increasing with latitude). By selecting only the observations which are made closest to the nadir condition, we ensure that observations are made at almost the same viewing conditions and, most importantly, observations are made at nearly the same local time globally for each level-2b product.

The level-2b approach leads to a significant reduction of the amount of used observations. However, the high observation frequency near the poles is undoubtedly very valuable and consequently there are also separate polar products added which are based on all available observations. The level-2b approach is used exclusively for cloud products and not for surface radiation and surface albedo products where the use of all existing data is more critical.

Final level 3 cloud products are available as daily and monthly composites where the monthly means are computed from daily means. Results are defined for each satellite on a regular latitude/longitude grid with a spatial resolution of $0.25° \times 0.25°$ degrees. In addition, results for cloud amount as well as the surface albedo (Section 5) are available on two equal-area polar grids at 25 km resolution for the Arctic and Antarctic regions, respectively; these grids are centred at the poles and cover areas of approximately 9000 km x 9000 km. The new features for CLARA-A2 include the availability of all daily level-2b products and a demonstration data record of probabilistic cloud masks (further explained in the next section).

Monthly averages of cloud products are also available in aggregated form (i.e., merging all satellites). Acknowledging the different observation capabilities during night and during day, and also taking existing diurnal variations in cloudiness into consideration, a further separation of data products into day and night has been performed. Here, all observations made under twilight conditions (solar zenith angles (SZA) between 75° - 95°) have been excluded in order to avoid being affected by specific cloud detection problems occurring in the twilight zone (e.g. Derrien and LeGleau, 2010).

All products described in the following three sections are described in detail in Product User Manuals (PUM), Algorithm Theoretical Basis Documents (ATBD) and Validation reports (VAL), all available via the CM SAF web user interface (accessible from www.cmsaf.eu). These documents are important as they describe and reference the latest algorithms utilized in the processing of the CLARA-A2 data record; the peer-reviewed



publications of retrieval algorithms referred to in the following Sections 4-6 may not always be up-to-date with
these very latest algorithm changes.





## 4    Cloud products

A list of all CLARA-A2 cloud products is given in Table 1. Basic methods for deriving these parameters were
already introduced by Karlsson et al. (2013). Consequently, the following sub-sections only provide a brief
introduction to the products with the focus on describing the most significant improvements since CLARA-A1
and on introducing some new features of the data record.

### 4.1    Improvements of basic cloud products derived from the NWCSAF cloud processing package

The Cloud Fractional Cover (CFC) product is derived directly from results of a cloud screening, or cloud
masking, method. CFC for one particular instantaneous observation is defined as the fraction of cloudy pixels
per grid box compared to the total number of analysed pixels in the grid box, expressed in percent. This product
is calculated using the NWCSAF Polar Platform System (PPS) cloud processing software. PPS was first
introduced by Dybbroe et al.) (2005) but the software has undergone several upgrades since then. The PPS
method also computes the Cloud Top level (CTO) product, providing the cloud top level as geometrical height,
cloud top pressure or cloud top temperature. The CTO retrieval is using two different radiance matching
methods, one for clouds identified as opaque and one for semi-transparent clouds.
CLARA-A2 takes advantage of some significant upgrades of the cloud masking and CTO retrievals in the latest
PPS version . Regarding improvements of cloud masking, the utilisation of reference measurements from the
CALIPSO-CALIOP sensor (Winker et al., 2009, Vaughan et al., 2009) has been fundamental for the
development and validation of the methods, following approaches by Karlsson and Dybbroe (2010) and
Karlsson and Johansson (2013). PPS dynamic cloud masking thresholds have been adjusted to detect a larger
fraction of the thinnest clouds and to better account for variations in surface emissivities/reflectivities in arid and
semi-arid regions. The challenging cloud screening conditions near the poles have also received special
attention. Cloud detection during Polar day conditions over snow- and ice-covered surfaces has been optimised,
and falsely-detected clouds during Polar night conditions have been largely removed. The latter unfortunately
leads to a systematic underestimation of cloudiness over the Arctic and Antarctica during the Polar night.
However, this better reflects the cloud detection limitations of the AVHRR sensor in situations with cold ground
temperatures than for the previous case with spuriously occurring false cloudiness in cold situations.
Figure 2 compares results from CLARA-A1 and CLARA-A2 using global, synoptic surface observations
(SYNOP) of cloud cover. For this study, the CLARA-A2 monthly mean product, generated from all available
satellites, was compared against SYNOP monthly mean cloud cover calculated based on daily means. Only those
stations and months where at least 6 observations per day for 20 days of the respective month were included in
the comparison (see the VAL report for more details). Results show relatively small changes in CFC bias but a
substantial decrease in the bias-corrected root mean squared error (bc-RMSE) for CLARA-A2. Thus, a much
better agreement with the SYNOP observed variability in cloud cover is achieved. The relatively unchanged bias
reflects inherent and unavoidable differences in the viewing geometry for the two observation types.



Figures 3 and 4 demonstrate the achievements made in cloud detection efficiency in CLARA-A2 in much more
detail. Results are based on an extensive cloud product monitoring effort utilising near-simultaneous (i.e., within
3 minutes) observations from the CALIPSO-CALIOP sensor over almost ten years (2006-2015). Despite the
nadir-only observation capability of the CALIOP sensor compared to the wide-swath coverage from AVHRR, it
has been possible to collect a global picture of cloud detection efficiency by accumulating results over the
relatively long time period. Figure 3 shows the global overall frequency of correct cloudy and cloud-free
estimations (referred to as Hitrate). Results show a general global agreement in cloud screening well above 80 %
apart from over the poles and high-latitude land areas and over high mountainous terrain.  Decreased Hitrate is
also found over the dry sub-tropical regions but this is mainly attributed to geolocations mismatches and
CALIOP-observed clouds frequently present on the AVHRR GAC sub-pixel scale (less than 4 km). In other
words, true small-scale or fractional clouds may exist in any of the two inter-compared data records but not
always simultaneously in both because of the small sizes of cloud elements. More serious is the somewhat poor
results seen over regions where cold surface conditions may prevail for considerable portions of the year. Figure
4 exemplifies this by showing the probability of detecting cloudy conditions over the Arctic. Over the coldest
portions of Greenland and the inner Arctic, almost 50 % of the clouds remain undetected in CLARA-A2 during
the polar winter. On the other hand, cloud screening complications are reduced during the polar summer when
results are nearly as good as over any other region of Earth (excluding some highly elevated areas of Greenland).
Comparisons have also been made to the MODIS (Moderate Resolution Imaging Spectroradiometer) sensor
Collection 6 data record  (http://modis-atmos.gsfc.nasa.gov/products_C006update.html) from the Aqua satellite
(Figure 5).  Generally, we find a very good agreement between the two data records, both in the geographical
distribution and in the zonal averages, of global cloud conditions for the overlapping data records of 2002-2014.
There is a bias of about 5 % in cloud cover (MODIS higher) which is relatively constant over all latitudes
(Figure 5, lower panels). This increase in clouds in MODIS data is interpreted as representing the improvements
in spectral channel availability of the MODIS sensor in comparison to AVHRR. However, the very good
correlation with MODIS results is encouraging considering the availability of two more decades of results from
AVHRR.
CLARA-A2 also includes a demonstration data record of probabilistic cloud masking following Karlsson et al.
(2015), defined in the level 2b data record. The alternative formulation here provides a measure of uncertainty in
cloud masking for the user to consult, compared to the traditional binary cloud mask utilized when compiling
level 3 CFC products. The intention is to shift entirely to a probabilistic formulation in the third edition of
CLARA planned for release in 2021.
The CTO retrieval in CLARA-A2 has been subject to several minor modifications while retaining the same
principle methodology. However, the most significant improvement is related to an optimisation of the iterative
procedure leading to a substantial efficiency leap regarding the fraction of resulting valid retrievals. The previous
method in CLARA-A1 was not able to provide valid estimations for all semi-transparent clouds which resulted
in that only about 70 % of all cloudy cases got valid CTO retrievals.   The new PPS version in CLARA-A2
provides CTO estimations for more than 97 % of all cases. This is especially important for the joint cloud-
histogram product (JCH, see section 3.3) and its ability to reflect true climatological conditions.



**4.2    Cloud products derived from the CM SAF Cloud Physical Properties (CPP) package**
The CPP products include cloud thermodynamic phase (CPH), cloud optical thickness (COT), particle effective
radius (REF) and liquid/ice water path (LWP/IWP). CPH is determined from a cloud typing approach following
Pavolonis et al. (2005). This cloud type algorithm consists of a series of spectral tests applied to infrared
brightness temperatures. It has a nighttime branch as well as a daytime branch in which shortwave reflectances
are also considered. COT and REF are retrieved using the classical Nakajima and King (1990) approach, which
is based on the principle that cloud reflectance is mainly dependent on COT at a non-absorbing, visible
wavelength and on REF at an absorbing, near-infrared wavelength. In the CPP algorithm (Stengel et al., 2014;
Roebeling et al., 2006), the Doubling-Adding KNMI (DAK, De Haan et al., 1987; Stammes, 2001) radiative
transfer model (RTM) is used to simulate visible (0.6 μm) and near-infrared (1.6/3.7 μm) TOA reflectances as a
function of viewing geometry, COT, REF, and CPH. These simulated reflectances are stored in a look-up table
(LUT) and satellite-observed reflectances are matched to this LUT in an iterative manner, leading to the
derivation of COT and REF. These two parameters are then used to compute LWP and IWP, as in Stephens
(1978). Uncertainty estimates of the CPP products are also derived and provided.
Major updates compared to the CPP version applied for CLARA-A1 (Karlsson et al., 2013) include the
implementation of the new cloud phase algorithm in the NWC SAF PPS software package (first made in PPS
version 2012 and for the latest improvements in PPS version 2014), the generation of improved cloud reflectance
LUTs , and the inclusion of observational sea ice and numerical weather prediction (NWP) snow cover data to
better characterize the surface albedo. CPP is now integrated within the NWCSAF PPS cloud processing
package. It should be noted that, since CPP retrievals require reflectances from shortwave channels, CPP
products, besides CPH, are available exclusively during daytime.
Figure 6 shows the CPH, LWP and IWP products averaged over the 5-year period 2003-2007. Large scale
climatological characteristics of clouds are apparent, including the marine stratocumulus regions off the west
coasts of the continents, the Inter-Tropical Convergence Zone (ITCZ), consisting mainly of ice clouds, and the
mid-latitude cyclone tracks on both hemispheres. High cloud water path values over polar regions should be
largely attributed to inadequate retrievals over snow- and ice-covered surfaces, providing little contrast with
clouds in the AVHRR visible channel.
Inter-comparison efforts with other similar data records show generally agreement better than 5 % for CPH (i.e,
for absolute frequencies of water clouds) and 0.005 kg m$^{-2}$ for LWP and IWP, although the bias compared to
DARDAR IWP is larger. More details on these results can be found in the VAL report.
Further illustrations of LWP and IWP results are given in Figs 7 and 8. Figure 7 shows the monthly time series
of LWP in the Tropics from CLARA-A2, along with two other satellite-based data records. Between them,
PATMOS-x is the most similar to CLARA-A2, since it covers the same period and was based on the same
(AVHRR) measurements. MODIS, on the other hand, covers the last 12 years of the time series and is the most
stable, since it involves a single (here: MODIS on Aqua is used), well-calibrated instrument. In general, the LWP
records agree well in terms of seasonal variability (see also the bottom panel of Fig. 7) and absolute amount of



tropical LWP. Both CLARA-A2 and PATMOS-x show some trends during various parts of the time series,
which should be largely attributed to orbital drift. It should also be noted that during the period 01/2001-
05/2003, channel 3a of AVHRR onboard NOAA-16 was switched on and used for the retrievals, instead of
channel 3b, which was used throughout the rest of the time series. This switch causes a jump in the time series of
both CLARA-A2 and PATMOS-x. Comparisons of LWP were also made against an independent, microwave-
based data record (O'Dell et al., 2008), focusing on the main stratocumulus regions, where liquid clouds prevail
(not shown). Results showed good agreement in both the seasonal cycle and absolute values of LWP, with an
average bias of -0.0034 kg m$^{-2}$, fluctuating in the range ±0.01 kg m$^{-2}$. Furthermore, Fig. 8 shows a validation of
pixel-level CLARA-A2 IWP with Cloudsat-CALIOP-based DARDAR observations (Delanoë and Hogan, 2008).
An overall underestimation by CLARA-A2 is observed, which becomes larger at high IWP values. Further
analysis indicates that this disagreement is mainly caused by differences in REF (especially for thick clouds),
while COT agrees well between the two data records (not shown).

### 4.3 Multi-parameter cloud product representations

The joint cloud property histogram (JCH) product is a combined histogram of CTP and COT covering the
solution space of both parameters (e.g., Rossow and Schiffer, 1991). This two-dimensional histogram gives the
frequency of occurrences for specific COT and CTP combinations defined by a constant bin space, separable for
liquid and ice clouds. This product is defined on a slightly coarser grid (1°x1° resolution) in order to achieve
higher statistical significance and to maintain manageable file sizes. The product is currently archived on the
grid-point resolution, so user-defined JCH analysis regions can be created.
Since the JCH product is a product visualisation technique, its quality is dependent on the quality of the
visualized products, including CTO (here, cloud top pressure), COT and CPH. Improvements of those products
have already been described but we repeat some of the valid points here:
- The increase in the number of valid CTO results gives a better representation of the true CTP-COT
distribution
- The histograms are now based on cloud products defined in the level-2b representation mode giving a
more homogeneous and consistent data distribution
- Frequencies of occurrences in each bin as well as the total cloud cover for all cases are now given
(although the latter can still deviate slightly from the CFC product value since for JCH we require all
three products - CTO, COT and CPH – to be simultaneously available).
Figure 9 shows global CLARA-A2 JCHs for afternoon satellites together with corresponding results from Aqua-
MODIS Collection 6 and PATMOS-x over the period 2003-2014 (i.e., the Aqua-MODIS era). In comparison to
global JCH results for CLARA-A1 (Karlsson et al., 2013) we highlight that clouds are now more frequent at
higher and lower tropospheric levels. This agrees well with MODIS and PATMOS-x, although the latter two
have more boundary layer clouds present, especially over open water (Fig. 9d-i).
Over land, MODIS and PATMOS-x distributions show an increased frequency of mid- and high-level clouds,
and a reduction in shallow cumulus and stratiform clouds (Fig. 9f, i). A relative increase in very optically thick
mid- and upper-level clouds, representative of nimbostratus and deep convection, also emerges for MODIS and



PATMOS-x. CLARA-A2 distributions generally agree with these distribution changes, although with CLARA-
A2 there is a tendency to observe a higher frequency of optically thinner clouds (COT ranging 0.3-3.6) across
the tropospheric column (Fig. 9c). Furthermore, there is a substantial amount of very optically thick mid- to
upper-level clouds in CLARA-A2 and PATMOS-x (Fig. 9c, i), which are largely absent in MODIS (Fig. 9f). In
CLARA-A2, this feature is linked to problems in estimating COT properly over snow-covered surfaces and
therefore COT products over these surfaces should be treated with caution. A JCH where the Antarctic continent
was masked resulted in the removal of this relative peak of high COT at mid- to high cloud levels in CLARA-A2
(not shown).
## 5    The surface albedo product
The cloud mask and AVHRR radiance data have been used as primary input data to generate the CLARA-A2
surface albedo (SAL) product of terrestrial black-sky surface albedo (wavelengths of 0.25-2.5 μm). It is available
as pentad (five-day) and monthly means and has the same spatial resolution and projection(s) as the other
CLARA-A2 products. Examples of the CLARA-A2 SAL product for January and July 2012 are given in Fig. 10.
The retrieval algorithm of CLARA-A2 SAL follows the same outline as the previous CLARA-A1 SAL
described in detail in Riihelä et al. (2013): After cloud masking, the possible effect of topography on geolocation
and radiometry in locations with inclined slope is corrected. Then, for the pixels on land, a correction for
scattering and absorption effects of aerosols and other atmospheric constituents is performed. In CLARA-A2
SAL, a dynamic aerosol optical depth (AOD) time series has been used. It has been composed using the Total
Ozone Mapping Spectrometer (TOMS) and Ozone Monitoring Instrument (OMI) aerosol index data
(Jääskeläinen et al., 2016). A correction for reflectance anisotropy of vegetated surfaces and spectral albedo is
then calculated. In CLARA-A1, one land use classification (LUC) was used for the whole time series. For the
current SAL product, four different LUCs are used. Finally a narrow-to-broadband conversion is made to derive
the albedo over the full spectral range of the product (0.25-2.5 μm). Since the reflectance anisotropy of snow is
large and varies according to snow type (Peltoniemi et al., 2005), the albedo of snow and ice covered areas is
derived by averaging into pentad or monthly means the broadband bidirectional reflectances of the AVHRR
overpasses. These overpasses are found to cover the whole viewing hemisphere (SZA smaller than 70° and
satellite zenith angles smaller than 60°) in most of the cases, giving a good representation of the bidirectional
reflectance distribution function. For the observations over open water, the albedo is constructed as a function of
SZA and wind speed. Wind information is taken from microwave measurements (SMMR and SSM/I data) and
available SYNOP observations. The classification between open water and sea ice has been verified using the
Ocean and Sea Ice Satellite Application Facility (OSI SAF) sea ice extent data (Eastwood, 2014).
In summary, the main differences in the algorithm between CLARA-A1 SAL and CLARA-A2 SAL are as
follows:
-  atmospheric correction uses dynamic AOD time series
-  number of LUCs used has been increased from one to four
-  wind speed data is used over sea to describe the sea surface roughness





The data record has been validated against in situ albedo observations from the Baseline Surface Radiation
Network (Ohmura et al., 1998), the Greenland Climate Network (Steffen et al, 1996), and the TARA floating ice
camp (Gascard et al., 2008). The sites have been chosen according to data availability, temporal coverage of
measurements and quality of data. The validation results show that CLARA-A2 SAL has a relative accuracy of
10-20 % over vegetated sites, and typically 3-15 % over snow and ice. Larger differences between the in situ
measurements and the satellite-based albedo value are mostly related to the heterogeneity of high-resolution
near-infrared surface reflectances at CLARA-A2 SAL pixel scales. The spatial representativeness is an issue at
most of the sites and should always be considered when using measurements of different scale (and location) for
validation (Riihelä et al., 2012).
The SAL time series was also compared to MCD43C3, the surface albedo product from MODIS (Schaaf et al.,
2002). The comparison showed that on a global scale, the two products are in good agreement. An overview of
the MODIS comparison results can be seen in Figure 11, showing the mean black sky albedo. These data have
been averaged over the common retrievable land/snow area after coarsening the MODIS product to 0.25° spatial
resolution and averaging the CLARA-A2 SAL pentad means to fit the MODIS products (delivered as 16-day
means). Water areas are excluded from the analysis since the MODIS product is not defined for water bodies
(including sea ice areas). The two products are in good agreement and generally the albedo differences are less
than 5 %, especially during the latter half of 2009. In general, the difference is caused by the methodology
differences, where the MODIS albedo product is normalized to local noon, which, for surfaces other than snow,
produces the minimum daily albedo. Taking this into consideration, CLARA-A2 SAL values should be slightly
higher than the MODIS product values. An analysis of the differences on latitudinal bands (not shown) shows
that over the northern hemisphere, the largest differences appear over Arctic land areas. The topography (which
is corrected for in SAL but not in the MODIS product) also creates differences in the average albedo at
mountainous regions.
The temporal stability of the CLARA-A2 SAL time series has also been evaluated using the central part of the
Greenland ice sheet (not shown) as a site whose albedo is expected to remain fairly constant over a long period
(Riihelä et al., 2013). The results showed that the maximum deviation of monthly mean CLARA-A2 SAL over
this site from its 34-year mean was 8.5 %, including some natural variability associated with e.g. varying SZA.
Also, the 34-year mean albedo for this site was estimated to be 0.786 which is somewhat lower than the literature
citations for the albedo of dry fresh snow (0.85, Konzelmann and Ohmura, 1995).
The cloud mask used in CLARA-A2 SAL is less conservative than the one used in CLARA-A1 SAL. This is
likely to affect the SAL values especially during non-continuous cloud conditions. Also the inaccuracies in the
land cover data record used to resolve the CLARA-A2 SAL algorithms may cause retrieval errors. The users are
recommended to utilize the existing support data (for example number of observations, standard deviation, mean
SZA, skewness and kurtosis per pixel) to remove suspect retrievals from their analysis.
Our quality assessment of the CLARA-A1 SAL surface albedo data record has shown that SAL retrievals over
snow and ice, particularly over the Arctic, are of good quality (Riihelä et al., 2010). Also, according to user
feedback the data record has been useful for climate model validation (e.g., Light et al., 2015). The retrieval over
snow and ice is essentially the same in CLARA-A2 SAL as it was for the previous edition of the data record





which gives reason to believe that the user feedback and quality assessment should, to some extent, also be valid
for CLARA-A2. The validation results against in situ observations and comparison with MODIS MCD43C3
product show that adding the new AOD time series for land areas has improved the algorithm performance
elsewhere as well.
## 6    Surface radiation products
The retrieval algorithms to derive the CLARA-A2 surface radiation products have only undergone minor
changes since CLARA-A1. Details on the algorithms are given in Karlsson et al., (2013). Thus, this section
presents a few validation results of the CLARA-A2 surface radiation data records.
### 6.1    Surface Solar Irradiance
The spatial data coverage of the surface solar irradiance (SIS) data has been substantially improved. In CLARA-
A2, only snow-covered surfaces are excluded due to a reduced accuracy of the SIS data under these conditions.
The validation against surface reference measurements from the Baseline Surface Radiation Network (BSRN)
documents the improved accuracy of the CLARA A2 surface irradiance data record, mainly due to the improved
cloud detection (see Table 2).
Figure 12 presents the comparison of the decadal linear trends derived from the CLARA-A2 SIS data record
with the corresponding trends derived from measurements obtained from the BSRN. To assess the validity of the
linear trend derived from the CLARA data record, only surface stations with continuous observations covering at
least 10 years of measurements are used.
The trends derived from the CLARA-A2 surface irradiance record correspond well to the trends derived from the
BSRN measurements, indicating the high stability of the satellite-derived product and its suitability to calculate
temporal changes and trends (Fig. 12). For most BSRN stations, the decadal trend is positive during the
considered time period. Note that the time period for which BSRN measurements are available differs between
the stations; consistent time periods were used to compare the CLARA-A2 SIS data record with the BSRN
measurements at each station.
Figure 13 presents the spatial distribution of the decadal linear trend between 1992 and 2015 based on the
CLARA-A2 SIS data record in Europe and parts of Northern America. To limit the impact of the missing data
during the first decade of the CLARA-A2 SIS data record due to the availability of only one AVHRR
instrument, the trend was derived starting in 1992, when at least two AVHRR instruments have been available.
In both regions there is an overall positive trend in surface irradiance, consistent with surface observations (e.g.,
Wild, 2012).
### 6.2    Surface Longwave Radiation
The CM SAF CLARA-A2 data record provides information on the surface longwave downwelling (SDL) and
outgoing (SOL) radiation in order to enable studies of the full surface radiation budget. Both data records are
dependent upon the surface and TOA longwave radiation records from the ERA-Interim reanalysis (Dee et al.,



2009); using topographic information and the monthly mean cloud fraction from CLARA-A2, the ERA-Interim
data are downscaled to match the spatial resolution of the CLARA–A2 data record. For SOL, this means a pure
downscaling of ERA-Interim data. For SDL, an effective cloud factor is derived, based on ERA-Interim
differences in clear-sky and all-sky downwelling longwave radiation, and reanalysis cloud fraction (Karlsson et
al., 2013). This factor is then downscaled to CLARA-A2 resolution and multiplied by the CLARA-A2 satellite-
derived cloud fraction. The result is a hybrid estimate of combined satellite-reanalysis SDL.
Table 3 shows the validation results of the monthly mean CLARA-A2 SOL and SDL data records compared to
measurements obtained from the BSRN network. The improved cloud mask in CLARA-A2 led to a substantial
improvement of the data quality of SDL data record relative to the CLARA-A1 data record.
**7    Discussion: Demonstration of potential applications**
The improvements of the underlying AVHRR radiance data record, the upgraded retrieval methods and the 6-
year prolongation of the observation time series all increase the usefulness of the CLARA CDR for many
applications. In other words, the potential for the data record to quantify true climate variability and trends has
improved.
In this section, we provide two examples demonstrating different applications of the new CLARA-A2 CDR. The
demonstrated results should be viewed as preliminary results requiring further detailed analysis. The purpose of
presenting them here is meant to encourage in-depth follow-on studies.
The first example deals with the following question: How have surface and cloud conditions changed in the
Arctic region during the last three to four decades? Similar studies on Arctic surface albedo variations alone have
already been made based on CLARA-A1 data (Riihelä et al., 2013), but access to a longer time series of
observations (including the new record year 2012 in Arctic minimum sea ice extent) and the coupling to cloud
processes clearly motivate continued studies in this field.  Many climate predictions and scenarios point at the
existence of an Arctic amplification (Cohen et al., 2014) of the temperature rise due to several large positive
feedback effects; two of these effects are the decrease of sea ice cover and its interaction with cloud cover. The
good AVHRR observation conditions during the polar summer season (e.g., as pointed out in section 3.1
discussing Figure 4) now permit more in depth studies of these two aspects.
Figure 14 shows the mean change of SAL for the first decade of the CLARA-A2 period (1982-1991) compared
to the last decade (2006-2015) over the high-latitude Northern Hemisphere during the summer months. The
corresponding changes in mean cloud cover are shown in Fig. 15. We can clearly see the strong SAL signal
associated with Arctic sea ice decline since the 1980s, which is very evident in all months from May through
September. Corresponding changes in Arctic cloudiness (Fig. 15) appear not as equally systematic and well
depicted. This is not surprising since cloud conditions depend primarily on atmospheric circulation patterns.
However, we notice a tendency for increased cloud cover over the marginal ice zone and the new ice-free
regions of the inner Arctic in the months July-September while some decreases in cloud cover can be seen over
the remaining ice-covered parts (e.g. close to Greenland and the Canadian archipelago). This result, based on
long-term CLARA-A2 data, supports the findings by Devasthale et al. (2016) regarding a similar co-variability





between cloudiness and sea-ice concentration observed in the last decade. An interesting feature in April-May is
also the increase in cloud cover in the inner Arctic region while cloudiness appears to decrease outside of this
area. Further studies should investigate the significance of these patterns and the possible links to changes in
circulation and radiation conditions. For these purposes, the entire CLARA-A2 data record (i.e., including years
1992-2005) must be used.
Another application is related to studies of cloud-climate feedback processes which are known to explain a large
part of the uncertainty in climate scenarios from climate models. Norris et al. (2016) claim that we can already
now see changes in global cloud patterns from long-term satellite observations, which are supported by climate
model simulations. Specifically, the authors claim that there are signs of a decrease in mid-latitude cloudiness
indicating "a poleward expansion of the subtropical dry zone cloud minimum and a poleward retreat of the
storm-track cloud maximum", based on a combined analysis of data from the ISCCP (Rossow and Schiffer,
1999) and PATMOS-x (Heidinger et al., 2104) climate data records covering the period 1983-2009. Similar
changes in cloud patterns have been noticed in corresponding CMIP5 historical simulations (Norris et al., 2016).
Since CLARA-A2 is based on the same basic satellite radiance data record as PATMOS-x, and only differing in
cloud retrieval methods, we could potentially see the same cloud changes. Figure 16 shows the linear trend,
expressed as cloud fraction change over a 25-year period but now estimated from 34 years of CLARA-A2 data,
in comparison to only 27 years of data analysed by Norris et al. (2016). Comparisons with corresponding results
by Norris et al. (2016) reveal many similarities, especially in the changed cloud distribution at low and tropical
latitudes. It is apparent that in the latter part of the 34-year period we have had some dominance of strong El
Nino events (e.g. 1997-1998 and 2015-2016) explaining the typical El Nino pattern over the central Pacific and
Indonesia/Australia regions in Fig. 16. Interestingly, this signal is not as strong as in the study by Norris et al.
(2016) which shows the value of studying a longer period where the ENSO extremes have a better chance of
neutralizing each other in the long-term mean.
We notice that changes in cloud distribution changes in other regions (e.g. the positive trends over Africa and in
the eastern Pacific region) cannot be as clearly linked to typical El Nino Southern Oscillation (ENSO) patterns
which is interesting. Whereas the agreement with Norris et al., (2016) is very good at low and tropical latitudes,
the changes seen at mid- and high latitudes in the CLARA-A2 results are not as conclusive. There are some
dominant negative trends over the typical extra-tropical storm track regions over the northern parts of the
Atlantic and Pacific oceans but a similar change over the southern oceanic regions is not evident. We also see
discrepancies over the eastern part of the Eurasian continent where CLARA-A2 shows a dominant decreasing
trend. Based on these results we find it difficult to conclusively support the suggestions by Norris et al. (2016)
regarding cloud changes at mid- and high-latitudes. If we consider the cloud trends derived from PATMOS-x
and ISCCP separately (conclusions were drawn from a composited ISCCP-PATMOS-x data record), which were
also provided by Norris et al. (2016), we find that mid-latitude trends for PATMOS-x show a better agreement
with CLARA-A2 trends. This highlights the strength of data records comprised from polar-orbiting satellites,
where better coverage of mid- and high-latitude regions is possible. Further studies are needed here to
understand these differences.



A final remark regarding the use of CLARA-A2 results in climate model evaluation studies is that a COSP cloud
observation simulator (Bodas-Salcedo et al., 2011) for CLARA-A2 is under preparation for facilitating
comparisons with satellite-based results. In that way artefacts in the satellite observation and adequate
corrections for viewing and observation conditions can be applied in order to give a more realistic inter-
comparison of results between models and CLARA-A2 cloud products.
**8     Summary and future plans**
We have described the CLARA-A2 dataset – an improved 34-year cloud, surface albedo and radiation budget
data record based on data from the AVHRR sensor on polar orbiting operational meteorological satellites. Major
improvements in both the underlying AVHRR radiances and in the retrieval schemes have been described,
together with some validation results. Regarding the latter, we have selected a limited glimpse at the exhaustive
results created through the extensive validation efforts that have been conducted. More results and analyses are
planned in follow-on papers. Some typical applications have also been demonstrated to encourage such studies
using CLARA-A2 data records. We would also like to highlight the broadening of the CLARA portfolio of
products which now also include daily aggregated and resampled orbits (level 2b) and the existence of an
experimental data record on probabilistic cloud masks. Related to this is also the development of a CLARA-A2
cloud dataset COSP simulator.
A continuation of this work has recently been secured by the EUMETSAT approval of the third continuous
operations and development phase (CDOP-3) of the CM SAF project covering the years 2017-2022.  This means
that a third edition of CLARA (CLARA-A3) is planned for release by the end of the CDOP-3 phase. This would
be the last edition based entirely on original AVHRR data, including data from METOP-C (the last polar satellite
carrying the AVHRR instrument).  Furthermore, it will include an extension of the dataset with data forward in
time for the years 2016-2020 and backward in time to 1978 (including data from the AVHRR/1 sensor starting
with the Tiros-N satellite), which means it will cover more than 40 years in time. The product dataset will then
also be extended with top of atmosphere radiation products and the original AVHRR radiances (level 1) will take
advantage of a revised infrared calibration (following Mittaz and Harris, 2009), in addition to the upgraded
visible calibration.





1    **Appendix A: Acronym list**

| | | |
|---|---|---|
| 2 | ACP | Atmospheric Chemistry and Physics journal |
| 3 | AOD | Aerosol Optical Depth |
| 4 | ATBD | Algorithm Theoretical Basis Document |
| 5 | AVHRR | Advanced Very High Resolution Radiometer (NOAA) |
| 6 | BSRN | Baseline Surface Radiation Network |
| 7 | CALIOP | Cloud-Aerosol Lidar with Orthogonal Polarisation (CALIPSO) |
| 8 9 | CALIPSO | Cloud-Aerosol Lidar and Infrared Pathfinder Satellite Observation satellite (NASA) |
| 10 | CDR | Climate Data Record |
| 11 | CERES | Clouds and the Earth's Radiant Energy System (NASA) |
| 12 | CFC | Cloud Fractional Cover product |
| 13 | CFMIP | Cloud Feedback Model Intercomparison Project |
| 14 15 | CLARA-A | The CM SAF cLoud, Albedo and surface RAdiation dataset from AVHRR data |
| 16 | CM SAF | Climate Monitoring Satellite Application Facility (EUMETSAT) |
| 17 | COSP | CFMIP Observation Simulation Package |
| 18 | COT | Cloud Optical Thickness product |
| 19 | CPH | Cloud Phase product |
| 20 | CPP | Cloud Physical Products package |
| 21 | CTO | Cloud TOp level product |
| 22 | DAK | Doubling-Adding KNMI radiative transfer model |
| 23 | ECMWF | European Centre for Medium-range Weather Forecasts |
| 24 | ENSO | El Nino Southern Oscillation |
| 25 | ERA-Interim | ECMWF ReAnalysis Interim dataset |
| 26 27 | EUMETSAT | EUropean organisation for exploitation of METeorological SATellites |
| 28 | FCDR | Fundamental Climate Data Record |
| 29 | GAC | Global Area Coverage (AVHRR, 5 km global resolution) |
| 30 | GCOS | Global Climate Observing System (WMO) |



| 1 | GEWEX | Global Energy and Water cycle EXperiment |
|---|---|---|
| 2 | ISCCP | International Satellite Cloud Climatology Project |
| 3 | ITCZ | Inter-Tropical Convergence Zone |
| 4 | IWP | Ice Water Path product |
| 5 | JCH | Joint Cloud property Histograms |
| 6 | LUC | Land Use Classification |
| 7 | LUT | Look Up Table |
| 8 | LWP | Liquid Water Path product |
| 9 | MCD43C3 | MODerate-resolution Imaging Spectroradiometer (MODIS) Albedo product |
| 10 | MODIS | Moderate Resolution Imaging Spectroradiometer (NASA) |
| 11 | NASA | National Aeronautics and Space Administration (USA) |
| 12 | NOAA | National Oceanographic and Atmospheric Administration (USA) |
| 13 | NWCSAF | Nowcasting Satellite Application Facility (EUMETSAT) |
| 14 | PATMOS-x | The AVHRR Pathfinder Atmospheres Extended dataset (NOAA) |
| 15 | PPS | Polar Platform Systems package (EUMETSAT, NWCSAF) |
| 16 | PUM | Product User Manual |
| 17 | PyGAC | Python module for AVHRR GAC pre-processing |
| 18 | REFF | Cloud Effective Radius product |
| 19 | RTM | Radiative Transfer Model |
| 20 | SAL | Surface ALbedo product |
| 21 | SMMR | Scanning Multichannel Microwave Radiometer (Nimbus 7 satellite) |
| 22 | SSM/I | Special Sensor Microwave Imager |
| 23 | | (Defense Meteorological Satellite Program – DMSP – satellites) |
| 24 | SYNOP | Synoptical weather observations from surface stations |
| 25 | TOA | Top Of Atmosphere |
| 26 | VAL | VALidation report |
| 27 | WCRP | World Climate Research Programme |
| 28 | WMO | World Meteorological Organisation |
| 29 | | |
| 30 | | |





**Acknowledgements**

The authors want to thank Dr. Andrew Heidinger at NOAA for providing the method for inter-calibrating historic visible AVHRR radiances.

This work is funded by EUMETSAT in cooperation with the national meteorological institutes of Germany, Sweden, Finland, the Netherlands, Belgium, Switzerland and United Kingdom.

The CLARA-A2 data record is (as all CM SAF CDRs) freely available via the website https://www.cmsaf.eu.



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



**Table 1: CLARA-A2 level 3 cloud products.**

| Product identifier | Product name | Description |
|---|---|---|
| CFC | Cloud Fractional Cover | Average fraction (%) of cloudy pixels in grid point. |
| CTO | Cloud Top level | Cloud top defined in one of three options: geometrical height (m), pressure (hPa) or brightness temperature (K). |
| CPH | Cloud phase | Average fraction (%) of liquid water cloud pixels relative to all cloudy pixels. Product defined both day and night (new feature compared to CLARA-A1). |
| COT | Cloud Optical Thickness | Average (both linear and logarithmic) of cloud optical thickness for liquid and ice clouds separately (dimensionless). |
| REF | Cloud Effective Radius | Average of cloud particle sizes for liquid and icd clouds separately (μm). |
| LWP | Liquid Water Path | Average (in-cloud or all-sky) of column integrated liquid water (kgm$^{-2}$). |
| IWP | Ice Water Path | Average (in-cloud or all-sky) of column integrated frozen water (kgm$^{-2}$). |
| JCH | Joint Cloud Histogram | 2-D histograms of occurrences in predefined cloud top pressure – cloud optical thickness bins. Defined for both water and ice clouds in a 1° x 1° geographical grid. Only valid for daytime conditions (see text for further explanation). |



**Table 2: Validation results of the CLARA-A2 surface irradiance (SIS) data record (monthly mean / daily**
**mean) against the global data from the BSRN network; for reference the corresponding values for**
**CLARA-A1 are also given. Shown are the number of months/days, the bias and the absolute bias as well**
**as the correlation of the anomaly between the two CLARA data records and the BSRN data.**

| Data record | # obs | Bias (W/m$^2$) | Abs. bias (W/m$^2$) | Corr. Ano |
|---|---|---|---|---|
| CLARA-A2 | 6420 / 181649 | -1.6 / -1.7 | 8.8 / 27.7 | 0.87 / 0.90 |
| CLARA-A1 | 3105 / 96237 | -3.3 / - 4.7 | 10.4 / 34.3 | 0.88 / 0.85 |



**Table 3: Validation results of the monthly mean CLARA-A2 SOL and SDL data records compared to the**
**measurements from the BSRN network. As references also the results obtained from CLARA-A1 and**
**ERA-Interim are shown. Presented are the number of months used for the ccomparison, the bias and the**
**absolute bias, as well as the correlation of the anomalies.**

| Data record | # Months | Bias (W/m$^2$) | Abs. bias (W/m$^2$) | Corr. Ano |
|:---:|:---:|:---:|:---:|:---:|
| SOL <br>(A2/A1/ERA-I) | 1680 / 1270 / 1680 | 2.9 / 5.8 / 1.9 | 13.7 / 13.8 / 14.1 | 0.74 / 0.71 / 0.78 |
| SDL <br>(A2/A1/ERA-I) | 7302 / 5314 / 7302 | -4.7 / -3.7 / -6.4 | 7.9 / 8.3 / 9.4 | 0.84 / 0.82 / 0.84 |





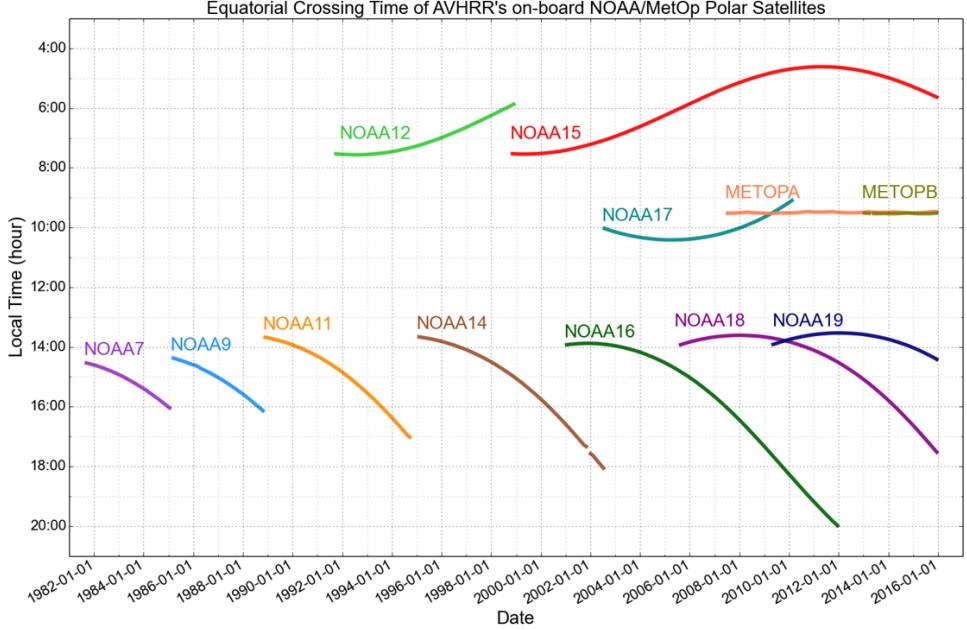

**Figure 1: Daytime equator observation times for all satellites covered by CLARA-A2 from NOAA-7 to NOAA-19 and METOP A/B. The figure shows ascending (northbound) equator crossing times for all afternoon satellites fromNOAA-7 to NOAA-19 and descending (southbound) equator crossing times for all morning satellites (NOAA-12, NOAA-15NOAA-17 and METOP A+B). Corresponding night-time or evening observations take place 12 hours earlier/later. Some data gaps are present but only for a number of isolated dates.**

11





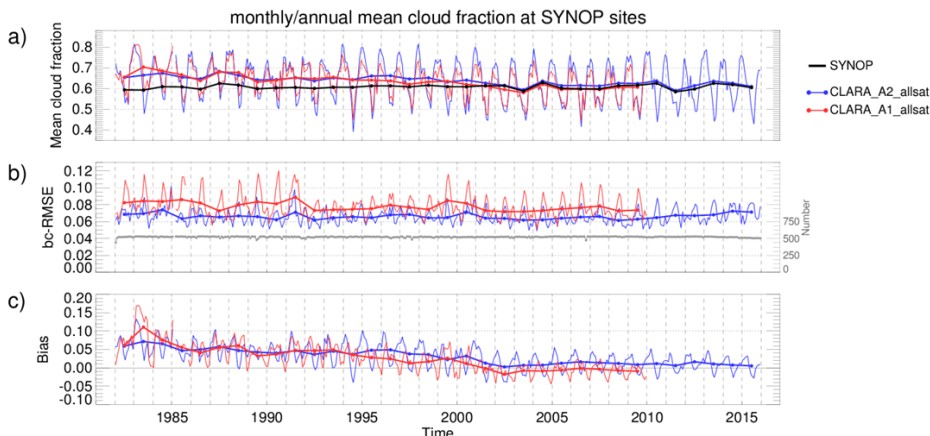

**Figure 2: (a) Time series of mean monthly and annual cloud fraction for CLARA-A2 (blue), CLARA-A1**

**(red), and SYNOP (black), (b) bias-corrected RMSE and (c) bias for the entire period 1982-2015. See text**

**for further details.**





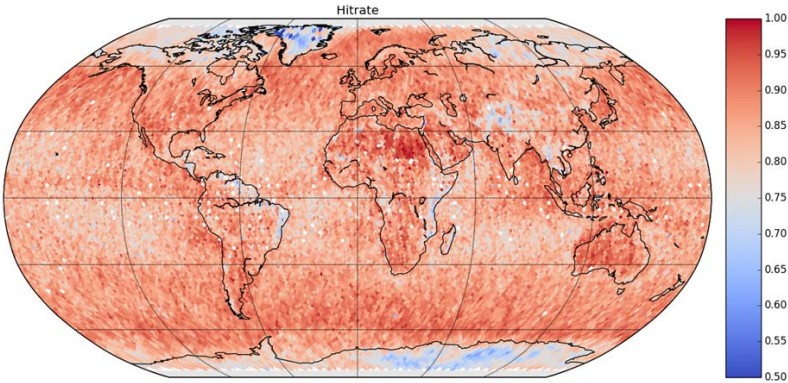

**Figure 3: Global overall frequency of correct cloudy and cloud-free estimations (often referred to as the Hitrate) derived from nearly 10 000 collocated (within 3 minutes) near-nadir AVHRR and CALIPSO-CALIOP orbits in the period 2006-2015. Results are collected in a Fibonacci grid with 28878 grid points evenly spread out around the Earth approximately 150 km apart. The resulting grid has almost equal area and almost equal shape of all grid cells. White spots are cells with insufficient coverage of collocations.**



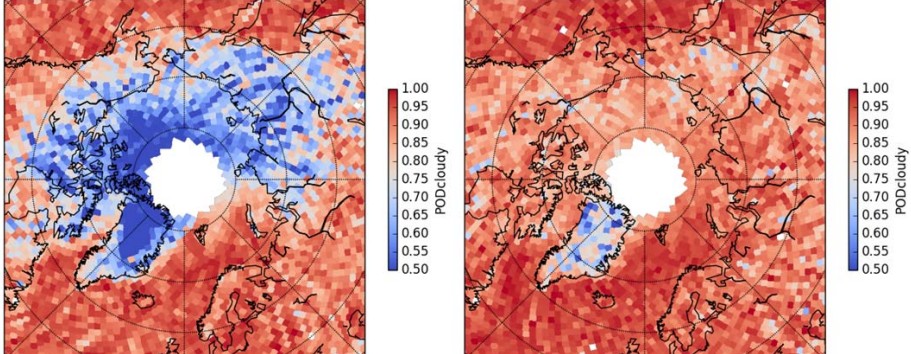

5  **Figure 4: Probability of detecting cloudy conditions over the Arctic region during the Polar Winter (left)**

6  **and during the Polar Summer (right). Results were derived from the same dataset as in Figure 3.**



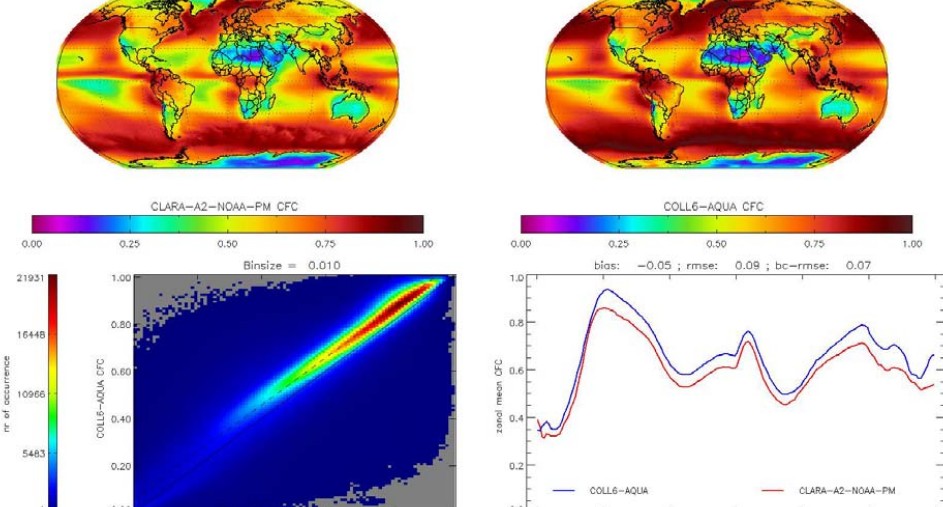

**Figure 5: Intercomparison of CLARA-A2 and MODIS Collection 6 (Aqua part) cloud fraction over the**

**covered MODIS period 2002-2014. Upper left: CLARA-A2 global cloud cover (CFC). Upper right:**

**MODIS global cloud cover. Lower left: Scatterplot of the two data records. Lower right: Latitudinal**

**distribution (zonal means) of cloud cover from the two data records (CLARA –A2 in red and MODIS in**

**blue).**



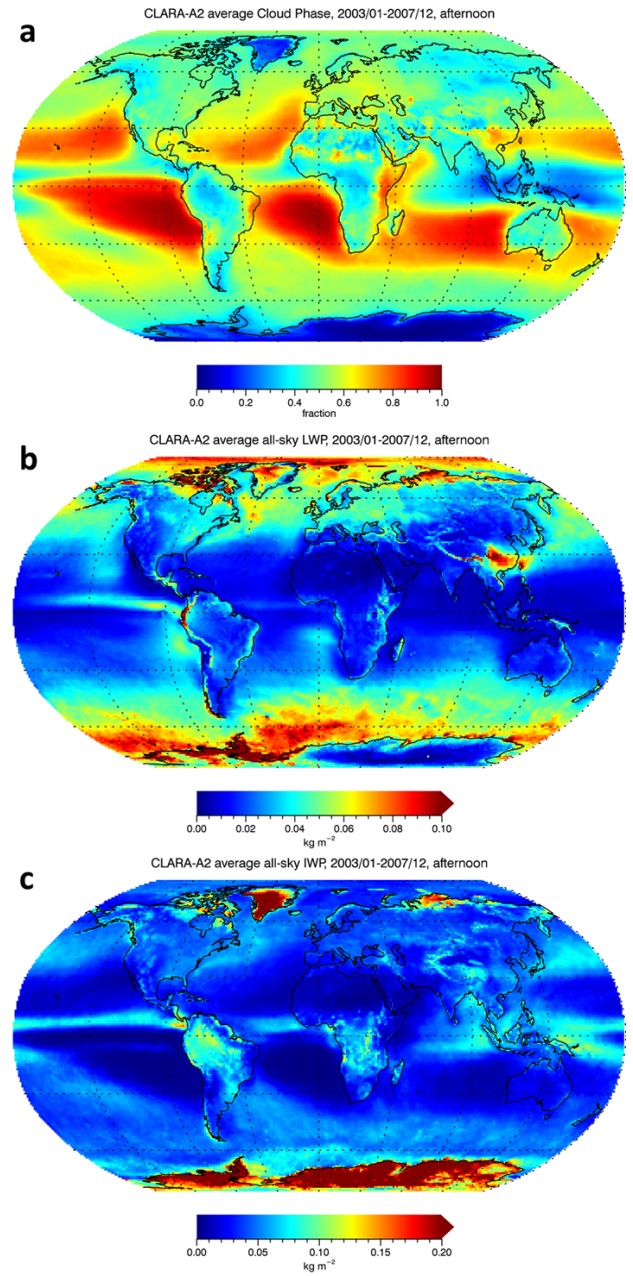

**Figure 6: (a) Fraction of liquid clouds relative to total cloud fraction, (b) all-sky liquid water path and (c)**
**all-sky ice water path, averaged for the 5-year period 2003-2007. All data come from CLARA-A2 level 3**
**products, derived from afternoon (NOAA-16 and NOAA-18) satellite measurements.**





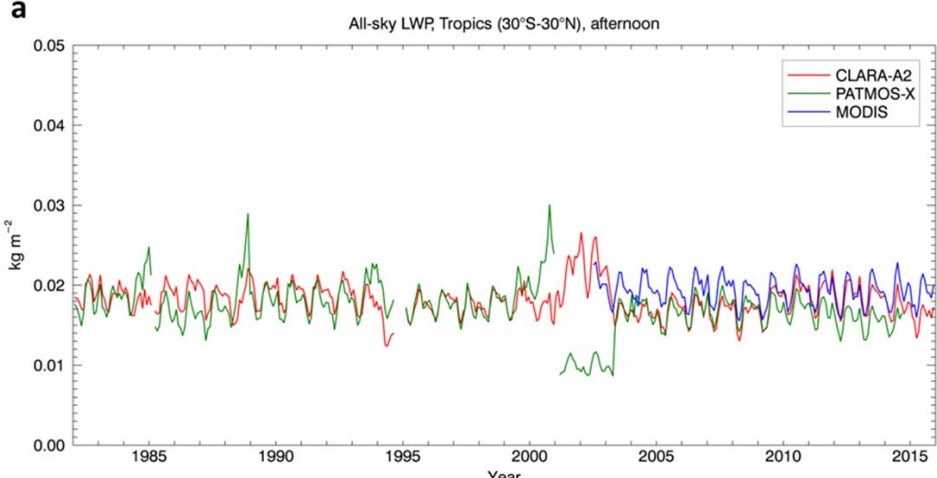

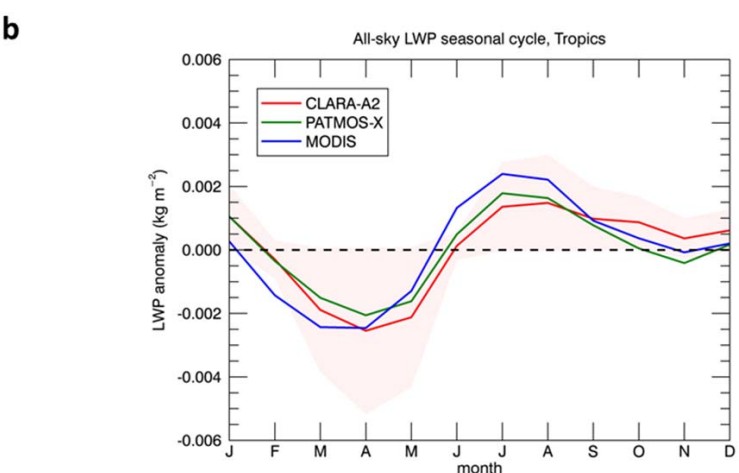

**Figure 7: Comparison between CLARA-A2, PATMOS-x and MODIS all-sky liquid water path (kgm$^{-2}$)**
**for the Tropics (30°S – 30°N): monthly time series (top) and seasonal anomaly (bottom). The seasonal**
**anomaly is calculated as the average of the deviations of monthly means from the corresponding yearly**
**mean over the years 2003-2013. The shaded area around the CLARA-A2 curve indicates +/- 1 standard**
**deviation of these deviations. The plots have been compiled from the NOAA afternoon satellites (NOAA-7,**
**-9, -11, -14, -16, -18, and -19) for CLARA-A2 and PATMOS-x and the MODIS Aqua (MYD08 Collection**
**6) 3.7 μm product.**


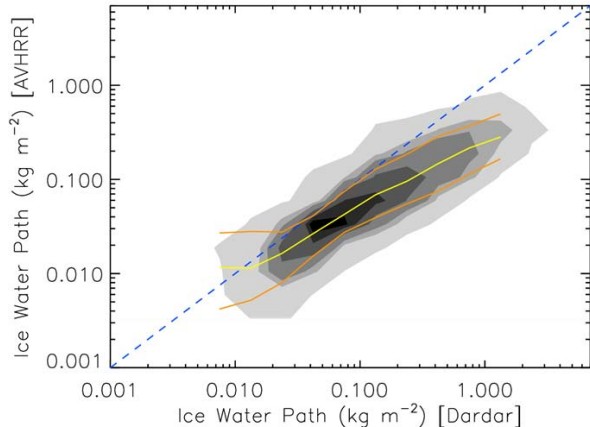

**Figure 8: CLARA-A2 (NOAA-18) IWP vs. DARDAR IWP for the months January and July 2008. The**

**yellow line depicts the median and orange lines the 16th/84th percentiles of the CLARA-A2 distribution at**

**the corresponding DARDAR IWP. The greyscales indicate regions enclosing the 10, 20, 40, 60, and 75% of**

**points with the highest occurrence frequency.**



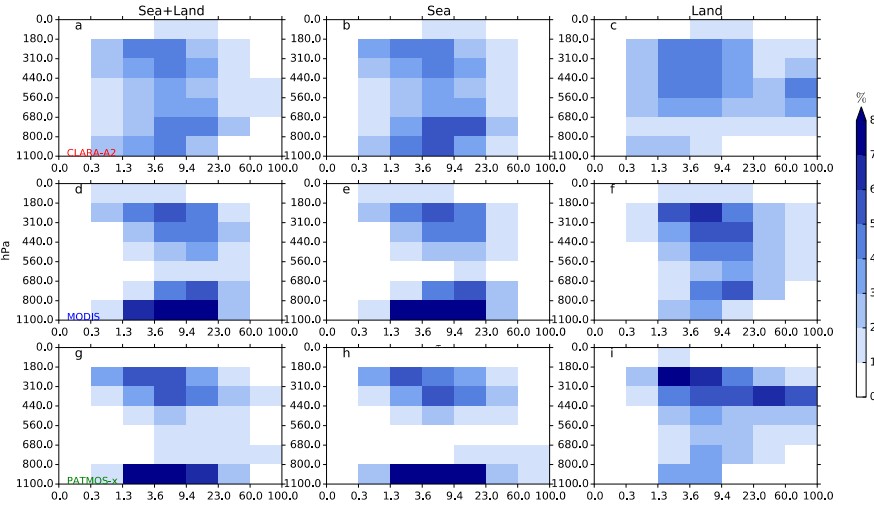

**Figure 9: Global JCH relative frequency distributions [colors, %] of CTP [hPa] and COT for all months during 2003-2014. The top row (panels a-c) are CLARA-A2, the middle row (panels d-f) are MODIS Collection 6, and the bottom row (panels g-i) are for PATMOS-x. Left column contains the JCHs over sea and land surfaces (sea+land), middle column over sea-only surfaces (sea) and right column over land-only surfaces (land). Histogram frequencies are here normalized to unity, such that each histogram sums to 100%.**



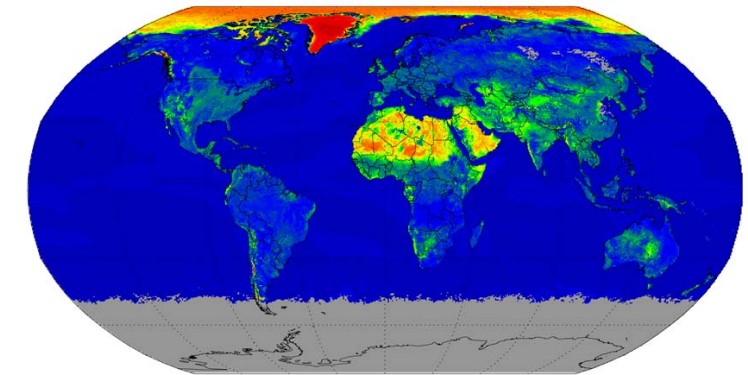

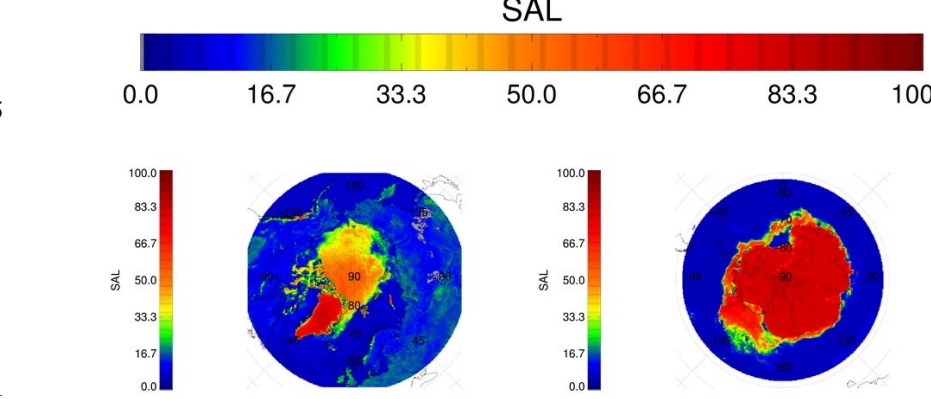

**Figure 10: Global monthly mean surface albedo for July 2012 (top). Corresponding plots for two polar grids are shown at the bottom of the figure; one for the Arctic region (bottom left) and one for the Antarctic region (bottom right, but observe that the month here is January instead of July). Regions without values are grey-shaded (here resulting from dark conditions prevailing close to Antarctica during the Polar winter).**

8
9
10






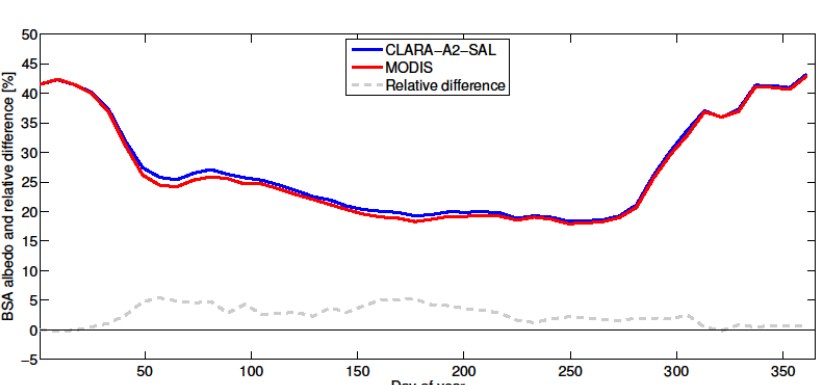

**Figure 11: Comparison of surface albedos from CLARA-A2 SAL (blue line) pentad composites and**
**MODIS MCD43C3 (green line) for 2009 [unit is per cent]. The means are calculated only over those**
**land/snow surfaces that are retrieved in both products, the MODIS product is not defined for water**
**bodies, thus they are excluded from this analysis. No weighing for irradiance or area has been applied.**
**The relative difference between the products is shown with a grey dash-dotted line.**





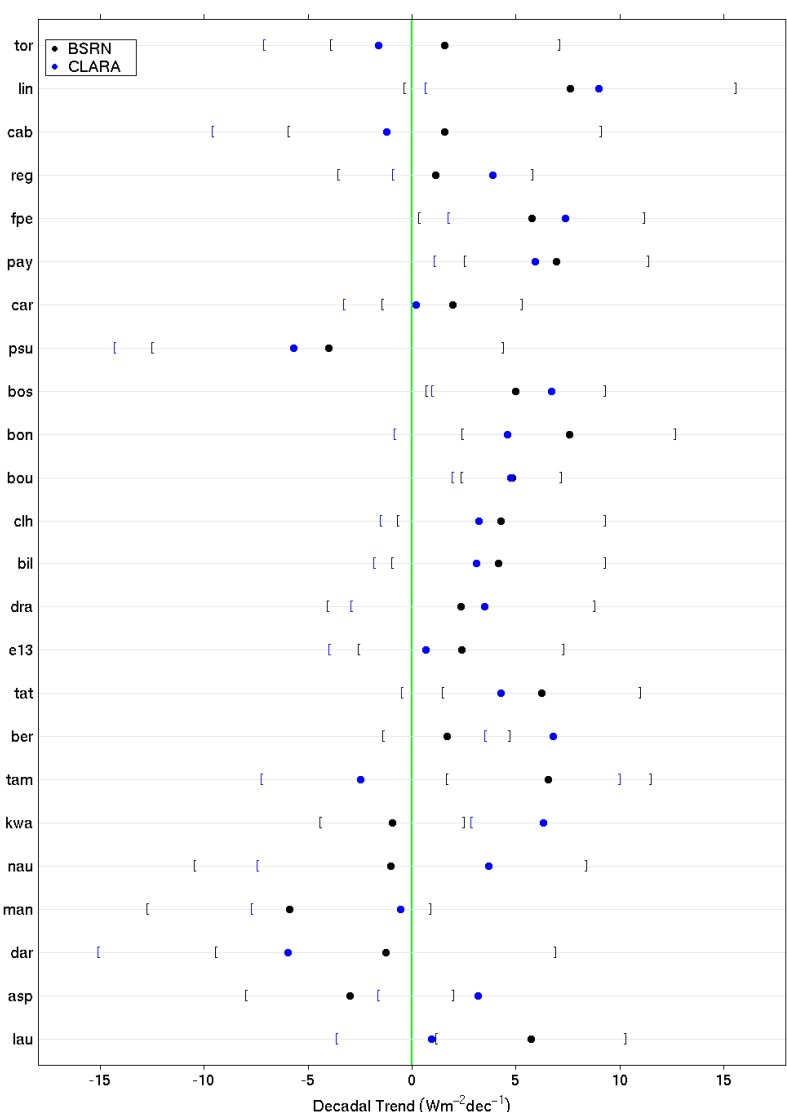

3 **Figure 12: Decadal linear trends derived from the CLARA-A2 surface irradiance data record (blue dots)**

4 **with the corresponding trends derived from measurements obtained from the BSRN (black dots). Trends**

5 **are shown as Wm$^{-2}$dec$^{-1}$. See text for further details.**


**a)**

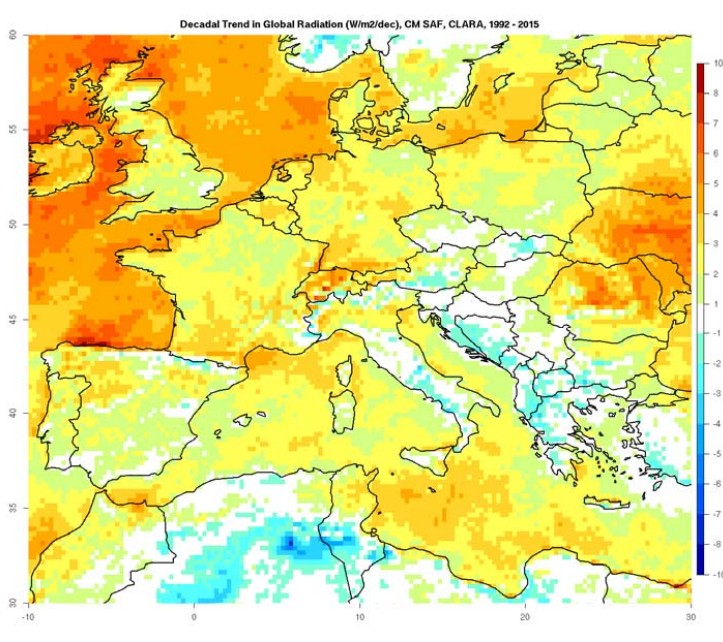

**b)**

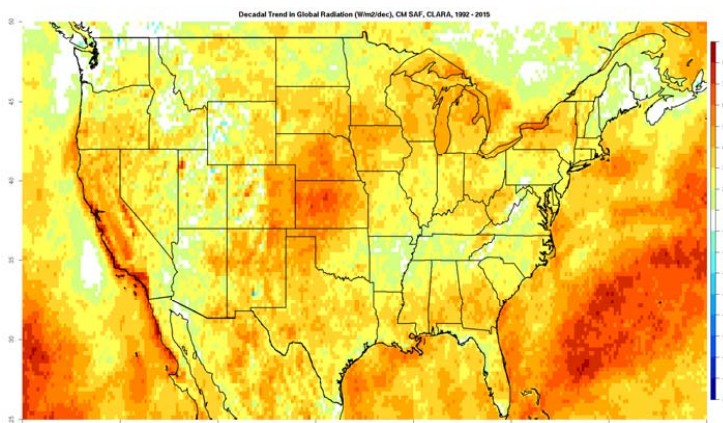

**Figure 13: Decadal linear trend of the surface irradiance from 1992 to 2015 based on the CLARA-A2 SIS**
**data record in (a) Central Europe and (b) parts of Northern America. All trends shown as Wm$^{-2}$dec$^{-1}$.**





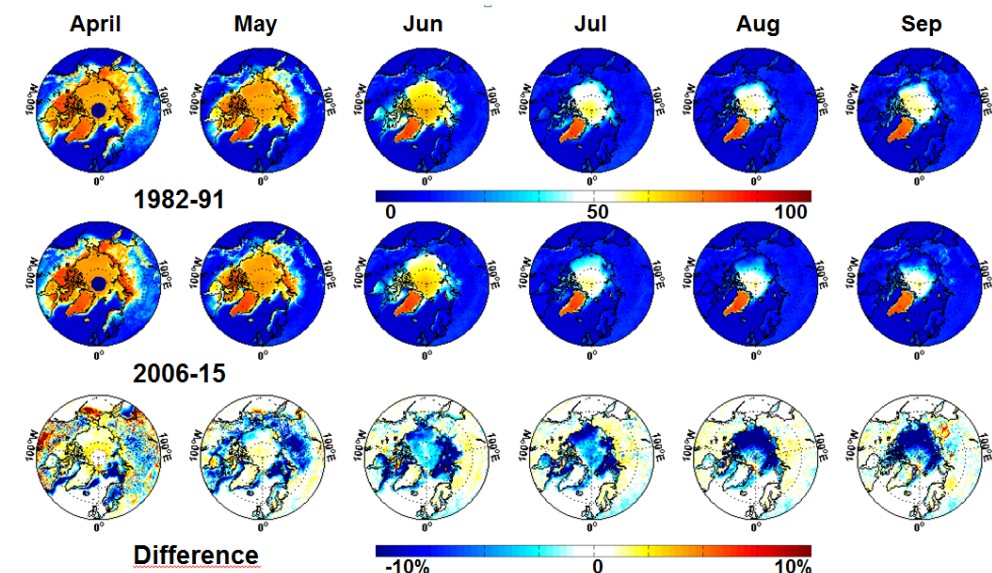

**Figure 14: Arctic summer season mean surface albedo for the first CLARA-A2 decade (1982-1991, upper**
**panel) compared to the last decade (2006-2015, middle panel). Difference plot (last minus first decade)**
**shown in the lower panel.**





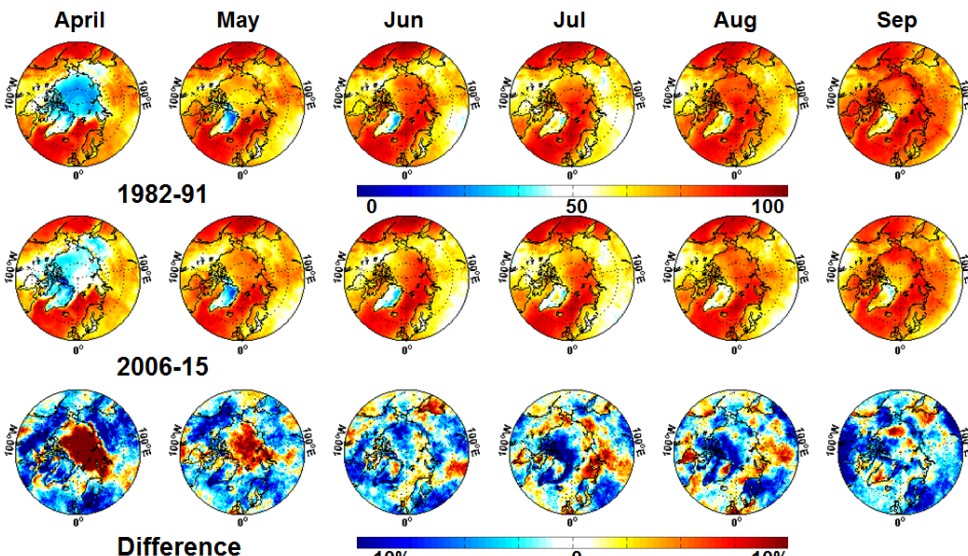

**Figure 15: Arctic summer season mean cloudiness for the first CLARA-A2 decade (1982-1991, upper panel) compared to the last decade (2006-2015, middle panel). Difference plot (last minus first decade) shown in the lower panel.**



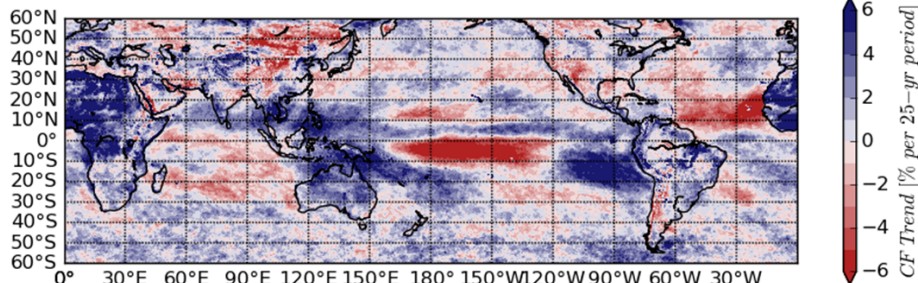

**Figure 16: Average 25-year trend in cloud cover (%) for low and middle latitudes calculated on the basis**
**of the average annual trend from CLARA-A2 in the period 1982-2015.**

