# Peer review of "CLARA-A2: The second edition of the CM SAF cloud and"

_Atmospheric Chemistry and Physics, 2016_

## Referee Comment (RC1) · M. Foster (Referee) · 12 Jan 2017

**General Comments**

The article is generally well written and organized. The subject matter represents a great deal of work and a significant contribution to the cloud climate community. That said there are ways I think the article could be improved. There is little discussion and few figures comparing CLARA-A1 to CLARA-A2, which I think makes it more difficult to understand the location and magnitude of improvements. I also think some of the changes could be described in more detail. Specific comments are below.

**Specific Comments**

[Figure]

Section 4 There are lists of specific tasks performed for GAC data pre-processing, histogram, and albedo products. The cloud masking changes are less clear. A list of specific changes might help this. For example, P5 L22 states "Cloud detection during Polar day conditions over snow- and ice-covered surfaces has been optimized, and falsely-detected clouds during Polar night conditions have been largely removed." How was this done? It might also help to show specific examples – a comparison scene of cloud detection over semi-arid regions for CLARA-A1 and CLARA-A2 would be one possibility.

Similarly there is little description of the changes to CTO retrieval. Would it be possible to include a little more detail as to what modifications were made to allow successful retrievals to jump from 70% to 97%?

There is not a lot of discussion of how the changes compare to CLARA-A1 (other than Figure 2). I think it would be helpful to include CLARA-A1 data in a few of the comparison figures against PATMOS-x and MODIS (and maybe have a Hitrate panel for CLARA-A1 in Figure 3). Figures 7, 9 and 11 seem like good candidates for this.

Section 7 The comparison against Norris et al. 2016 seems superficial, even by the preliminary standard defined in the manuscript. It is difficult to come to any conclusions based on the single figure 16. A linear regression to remove the ENSO signal might shed some light on this.

Figures 3 and 4 – the red-blue colorbar is usually used for temperature or something with positive and negative values. Also it is a little difficult to differentiate the value of Hitrate for higher values.

P4 – Are observations under twilight conditions excluded for all products, or just for the monthly averages?

P6 L7 – I don't understand this explanation. Is there perhaps a citation showing that the dry sub-tropical regions with decreased Hitrate are areas where sub-pixel scale

clouds frequently occur?

Figure 12 – Hard to differentiate between blue and black dots

Technical Corrections

P2 L10 – 'lined out' should be 'outlined' P2 L18 – 'already' is unnecessary and can be removed P2 L37 – the grammar and use of semicolon in this sentence is odd – consider rewording P3 L1 – Sentence beginning with "Additionally, orbital drift..." is awkward. Consider rewording P3 L9 – incorrect usage of the word 'spurious' P5 L13 "is using" should be "uses" P5 L26 – Should be "spurious" or "false" cloud, not both.

---

## Referee Comment (RC2) · Anonymous Referee #1 · 13 Jan 2017

The manuscript "CLARA-A2: The second edition of the CM SAF cloud and radiation data record from 34 years of global AVHRR data" by K.-G. Karlsspn et al. fits the scope of the journal and deserves to be considered for publication after some changes are made.

The paper is reasonably well written and understandable. Results and figures are provided with a sufficient quality.

Major comments

- The manuscript contains several statements on the improvements of CLARA-A2 with respect to the previous version CLARA-A1. While this is fine and informative for po-

tential users, such improvements are not sufficiently documented. Figure 2 presents a comparison, but there is not much more than this in the rest of the paper. The authors should consider adding some more material.

- Section 7 seems to me a bit too quick. The analyses presented in the section (chiefly the one on the change of cloud conditions in the Arctic and Northern Hemisphere) is admittedly conducted as a very preliminary effort. However, even adopting this perspective, the material and figures are integral parts of the paper and would require more studies in order to be published. I am not in favour of including this part. Perhaps the space saved could be effectively allocated to address the previous point of this review.

Minor comments

- The black/blue colour choice of the dots in the figure makes them difficult to spot. Maybe a black/red choice would do the job much better.

---

## Author Response (AR1)

[revised manuscript text omitted]

**Repeating general statement:**

The article is generally well written and organized. The subject matter represents a great deal of work and a significant contribution to the cloud climate community.

That said there are ways I think the article could be improved. There is little discussion and few figures comparing CLARA-A1 to CLARA-A2, which I think makes it more difficult to understand the location and magnitude of improvements. I also think some of the changes could be described in more detail. Specific comments are below.

**Reply:**

We thank the reviewer for this positive evaluation. In the reply to the specific comments we will outline how we plan to better illustrate the changes and improvements of CLARA-A2 compared to CLARA-A1.

In the following the specific review comments are commented and reference is made to the resulting changes in the revised manuscript (with line numbers taken from the Word document prepared in track change mode).

**Repeating specific comment 1:**

**Section 4: There are lists of specific tasks performed for GAC data pre-processing, histogram, and albedo products. The cloud masking changes are less clear. A list of specific changes might help this. For example, P5 L22 states "Cloud detection during Polar day conditions over snow- and ice-covered surfaces has been optimized, and falsely-detected clouds during Polar night conditions have been largely removed." How was this done? It might also help to show specific examples – a comparison scene of cloud detection over semi-arid regions for CLARA-A1 and CLARA-A2 would be one possibility.**

**Reply:**

- A list providing more details of the cloud masking algorithm changes will be added.
- For the specific question on cloud detection improvements in the polar regions we can say that we have systematically used CALIPSO cloud observations to identify when in particular falsely-detected clouds occur and with this information we have been able to reduce this problem considerably. For the polar night the focus has been on the latter rather than to improve polar cloud detection further since it is clear that AVHRR cloud detection capabilities during the polar night is already seriously limited. For daytime conditions the CALIPSO information has contributed to a better cloud discrimination over snow- and ice-covered surfaces. We will add this to the discussion in the text.
- Examples of changed results over semi-arid regions will be added.

**Final modification of manuscript:**

The text has been modified on **page 5 lines 17-36** and **page 6 lines 1-12** in the revised manuscript. A list has been added with more details on the algorithm improvements. Finally, a new Figure 2 has been added illustrating the changes in cloud amounts between CLARA-A2 and CLARA-A1 over the African continent, in particular highlighting the changes over semi-arid regions. In connection to this, two new references (Sun et al., 2015 and Sanchez-Lorenzo et al., 2017) have been added where the semi-arid problem for CLARA-A1 has been highlighted earlier.

Regarding the new Figure 2 one can see that the most remarkable changes are the removal of false-clouds over semi-arid areas (e.g. Sahel and parts of southern Africa) and an increase in the number of detected clouds over pure desert areas and vegetated areas. This leads to quite large changes in the cloud distribution over Africa. However, CALIPSO-based validation results in Figure 4 largely support the updated distribution given by CLARA-A2, thus we believe that we now have a much more realistic cloud distribution over Africa (and globally).

**Repeating specific comment 2:**

**Similarly there is little description of the changes to CTO retrieval. Would it be possible to include a little more detail as to what modifications were made to allow successful retrievals to jump from 70% to 97%?**

**Reply:**

- We will add more details. Basically, the improvement has resulted from applying more physically sound constraints to the iterations.

**Final modification of manuscript:**

Text has been modified on **page 8 lines 8-10**.

**Repeating specific comment 3:**

**There is not a lot of discussion of how the changes compare to CLARA-A1 (other than Figure 2). I think it would be helpful to include CLARA-A1 data in a few of the comparison figures against PATMOS-x and MODIS (and maybe have a Hitrate panel for CLARA-A1 in Figure 3). Figures 7, 9 and 11 seem like good candidates for this.**

**Reply:**

- We will consider adding more CLARA-A1 results, if possible. However, notice that no CLARA-A1 results have ever been produced for the period 2010-2015. Thus, since results in Figure 3 can only be visualised if having enough CALIPSO collocations (i.e., covering the full period 2006-2015) the specific request for that figure cannot be fulfilled. However, some inter-comparison results were produced for the period 2006-2009 and we can add this to the text (and a table).
- For Figures 7 and 11 we see no problem in also including CLARA-A1 results. Figure 9 needs some further consideration.

**Final modification of manuscript:**

Figures 7, 9 and 11 have now been complemented with CLARA-A1 results. However, notice that new figure numbers are Figure 8, Figure 10 and Figure 12 (which is a consequence of the inclusion of a new Figure 2).

Statements commenting these additional CLARA-A1 results can be found in the revised manuscript:

Figure 8: **page 9, lines 7-19.**

Figure 10: **page 10, lines 11-17.**

Figure 12: **page 11, lines 32-38 + page 12, lines 1-9.**

To notice is also the inclusion of a new Table 2 with inter-comparisons of CLARA-A1 and CLARA-A2 results for the same reference dataset from CALIPSO-CALIOP. Modified text about this can be found on **page 6 lines 21-30.**

**Repeating specific comment 4:**

**Section 7: The comparison against Norris et al. 2016 seems superficial, even by the preliminary standard defined in the manuscript. It is difficult to come to any conclusions based on the single figure 16. A linear regression to remove the ENSO signal might shed some light on this.**

**Reply:**

- We agree that results do not allow firm conclusions, which we also wrote clearly (this should only be seen as a demonstration of what can be studied using the data record). However, to claim that result would be superficial is a too strong statement. We just repeated the study (or parts of the study) in the Norris et al. 2016 paper to see what happens if we add another 7 years to the 27 years that they studied. There are obviously differences but also similarities so it is clear that it is difficult to find very clear conclusions. However, the indication that the cloud changes seen by Norris et al., 2016 for mid- and high-latitudes are maybe not as clear in our study despite having added several years (which should give better prospects of finding a long-term trend) would actually call for further and deepened studies here. For this reason we still think this addition to the paper is interesting.

  The recommendation to remove the ENSO signal by a linear regression is questionable. Firstly, this was not done by Norris et al. 2016 in the original study and, secondly, a possible climate change signal could actually also mean a changed behaviour of the frequency and amplitude of ENSO signals. Thus, it would be dangerous to assume a static ENSO behaviour.

  In conclusion, we still think that this section adds something to the

scientific discussion and that its presence could help trigger deeper studies about the specifically mentioned topics.

**Final modification of manuscript:**

We decided to keep Section 7 in the revised manuscript, even if this leads to a significant extension in the length of the paper (considering all other changes). The main reason is that we are keen on seeing more work being done on the exemplified applications so we really want to encourage this.

We removed the statement that results are preliminary and we instead underlined that the provided results are results directly derived from the CLARA-A2 data record (**page 14, lines 3-7).** We also emphasized again (**page 15, lines 4-8**) the importance and significance of having a longer time series of data (7 additional years) when repeating the study by Norris et al., (2016). However, we also repeated in both example cases that further in depth studies are needed here to evaluate the full impact of the results (**page 14, lines 29-31**).

See also discussion in final reply to Anonymous Referee 1.

Finally, we have also added the correct reference to the paper by Norris et al., (2016) which unfortunately was missing in the first version of the manuscript.

**Repeating specific comment 5:**

**Figures 3 and 4 – the red-blue colorbar is usually used for temperature or something with positive and negative values. Also it is a little difficult to differentiate the value of Hitrate for higher values.**

**Reply:**

We don't fully understand this comment. What we show is definitely something related to positive and negative values even if it expressed in a relative sense. Blue colours define poor validation scores and red values good validation scores. For example, in the case of Figure 4 the 50 % level of probability of detection must be considered as a critical negative case (i.e., here we only detect 50 % of all clouds). However, we admit that the choice of the intermediate point (i.e., when blue changes to red), which could be interpreted as the point where we go from bad to good results, is rather arbitrarily chosen. We will consider changing to a better color representation.

**Final modification of manuscript:**

We have introduced new color tables for the two figures (now renamed to Figure 4 and Figure 5). We basically avoided using the red and blue colours in this context.

Notice also the additional explanatory text about the used dataset for producing Figures 4 and 5 (**Page 6, lines 37-39 + Page 7, line 1).**

**Repeating specific comment 6:**

**P4 – Are observations under twilight conditions excluded for all products, or just for the monthly averages?**

**Reply:**

All cloud products except Cloud Physical Products (COT, REF, LWP, IWP) are based on all observations. The CPP exceptions are explained by the needed access to daytime visible channel data for the retrieval methods.

The exclusion of twilight data is restricted to complementary sub-layers to the cloud amount (CFC) and cloud phase (CPH) products. Thus, the main CFC and CPH products are based on all observations but in addition a user can choose to look also at CFC and CPH sub-layers showing results exclusively at daytime or exclusively at nighttime. When defining these two sub-layers no data under twilight conditions was used. These sub-layers are available for both daily and monthly CFC products. For the CPH product the standard product is complemented with a daytime product (no night-time product is prepared).

The description in the text is not correct and we will clarify.

**Final modification of manuscript:**

The text has been modified in three locations in the text:
1. **Page 4, lines 25-31** – introducing the idea of complementary day/night information.
2. **Page 5, line 13 –** specifying for CFC
3. **Page 8, lines 32-34)** – specifying for CPP and CPH

To repeat, the exclusion of twilight data only affects the CFC and CPH products yielding two complementary daytime and nighttime products in addition to the standard CFC product based on all data plus one complementary daytime CPH product.

**Repeating specific comment 7:**

**P6 L7 – I don't understand this explanation. Is there perhaps a citation showing that the dry sub-tropical regions with decreased Hitrate are areas where sub-pixel scale clouds frequently occur?**

**Reply:**

We are not certain what the problem is here (the reference to Page 6 Line 7 is not very specific). But if the question is only about the statement on the low Hitrate for dry-subtropical regions we can say the following:

Marine stratocumulus and cumulus clouds are dominant clouds over most marine ocean surfaces in the tropics and in the sub-tropics (if not being too close to the ITCZ). The frequency and the extent of clouds in these regions have definitely links to the size of the clouds. In the centre of sub-tropical anticyclones or highs cumulus clouds are mostly occurring as individual small-scale clouds (cumulus humilis +cumulus mediocris + cumulus congestus) with limited horizontal and vertical extent and with low frequency. Many of those cloud elements have sizes significantly smaller than the AVHRR GAC pixel (e.g., cumulus humilis or broken stratocumulus). However, away from the centre the number of clouds and their extent normally increases gradually with the distance from the centre. At some point the dominant cloud type may also change from individual cumulus clouds to stratocumulus clouds with larger horizontal extensions. We also often see a transition from cumulus clouds in open cell formation to closed cell formation. The cloud distribution is also affected by ocean current effects so that regions with colder ocean surfaces may lead to almost overcast stratocumulus conditions. Good examples here are the ocean waters outside (to the west of) Namibia and Peru. What we claim here is that, since the occurrence of really small and exclusive (i.e., not accompanied by larger scale clouds) cumulus cloud elements is more likely for the reasons explained earlier in the central regions of the sub-tropical highs, the risk of encountering matchup problems between AVHRR GAC pixels (with 5 km dimensions) and CALIPSO observations (with 300 m width FOVs) is higher here than outside of the central portions of the sub-tropical highs. We think that this is supported by the pattern in the Hitrate plots which highlights the decrease in Hitrate over typical positions of the sub-tropical highs. In conclusion, we believe that the reduced scores over sub-tropical high regions must be related to a higher relative frequency of small (sub-pixel scale) cloud elements among all

existing clouds leading to both enhanced CALIPSO collocation problems and to some extent also to a less efficient cloud detection.

The main problem here is that, neither the actual AVHRR GAC measurement (formed by sub-sampling only every fourth original 1 km resolution AVHRR scanline), nor the CALIOP observation (a narrow 330 m sampling in 15 measurements/lidar shots), is capable of covering the nominal GAC FOV more than to 10-20 %. Furthermore, they observe different portions of the GAC FOV since the AVHRR and CALIOP observation arrays are almost perpendicular. So when cloud element sizes go below 5 km the likelihood increases that the cloud is not observed simultaneously anymore in the two datasets. This explains the decreased validation scores.

The same thing could also happen over land areas with a high frequency of small-scale cloud elements but we think that the existence of more vigorous and widespread convection over land areas (as an effect of more heterogeneous surface conditions) might reduce this effect. However, we notice also low values over the eastern part of South-America and in eastern Africa which also could be linked to a high frequency of small cumulus cloudiness during periods of higher atmospheric stability.

To really prove this hypothesis is difficult (requires extensive high-resolution measurements over long periods) so we suggest that we modify the text in a way that we express this as a possible explanation rather than as a well-established truth. Maybe reliable global cloud-size statistics can eventually be collected from CALIPSO observations to reveal the answer, given that we could possibly be given a few more years of CALIPSO satellite operations.

**Final modification of manuscript:**

We have added a few clarifying sentences in the text (**Page 7, lines 1-17**) based on the more extensive reply above.

**Repeating specific comment 8:**

**Figure 12 – Hard to differentiate between blue and black dots**

**Reply:** We will try to improve the visibility here.

**Final modification of manuscript:**

The figure has been updated (new name is Figure 13). Dots have been made bigger and the blue colour has been changed to red.

**Technical Corrections:**

**P2 L10 – 'lined out' should be 'outlined' P2 L18 – 'already' is unnecessary and can be removed P2 L37 – the grammar and use of semicolon in this sentence is odd – consider rewording P3 L1 – Sentence beginning with "Additionally, orbital drift..." is awkward. Consider rewording P3 L9 – incorrect usage of the word 'spurious' P5 L13 "is using" should be "uses" P5 L26 – Should be "spurious" or "false" cloud, not both.**

**Reply:**

We will certainly correct this. Thanks for the suggestions.

**Final modification of manuscript:**

All technical corrections are implemented.

**Final reply to Referee 1's review of the ACPD paper**

**" CLARA-A2: The second edition of the CM SAF cloud and radiation data record from 34 years of global AVHRR data"**

**by**

**Karl-Göran Karlsson et al.**

**Repeating general statement:**

**The manuscript "CLARA-A2: The second edition of the CM SAF cloud and radiation data record from 34 years of global AVHRR data" by K.-G. Karlsson et al. fits the scope of the journal and deserves to be considered for publication after some changes are made.**

**The paper is reasonably well written and understandable. Results and figures are provided with a sufficient quality.**

**Reply:**

We thank the reviewer for this positive evaluation. We will reply to the specific comments below.

In the following the specific review comments are commented and reference is made to the resulting changes in the revised manuscript (with line numbers taken from the Word document prepared in track change mode).

**Repeating specific comment 1:**

**The manuscript contains several statements on the improvements of CLARA-A2 with respect to the previous version CLARA-A1. While this is fine and informative for potential users, such improvements are not sufficiently documented. Figure 2 presents a comparison, but there is not much more than this in the rest of the paper. The authors should consider adding some more material.**

**Reply:**

Yes, we will present more inter-comparisons with CLARA-A1 results. For example, we did cloud product inter-comparisons based on CALIPSO observations for both data records for the period 2006-2009 which were not reported in the manuscript. We will add some of these results. Also for other CLARA-A2 parameters we will try to add more inter-comparisons against CLARA-A1. For example, Figures 7 and 11 will be updated with CLARA-A1 results.

**Final modification of manuscript:**

A new table (Table 2) has been added showing inter-comparisons of CLARA-A1 and CLARA-A2 results being compared to a reference dataset of CALIPSO observations. Corresponding text in the manuscript is found on **page 6, lines 21-30**.

Figures 7, 9 and 11 have now been complemented with CLARA-A1 results. However, notice that new figure numbers are Figure 8, Figure 10 and Figure 12 (which is a consequence of the inclusion of a new Figure 2).

Statements commenting these additional CLARA-A1 results can be found in the revised manuscript:

Figure 8: **page 9, lines 7-19.**

Figure 10:  **page 10, lines 11-17.**

Figure 12:  **page 11, lines 32-38 + page 12, lines 1-9.**

The text has also been modified further (**page 5, lines 17-36 + page 6, lines 1-12**) in the revised manuscript. A list has been added with more details on the algorithm improvements since CLARA-A1. Finally, a new Figure 2 has been added illustrating the changes in cloud amounts between CLARA-A2 and CLARA-A1 over the African continent, in particular highlighting the changes over semi-arid regions. In connection to this, two new references (Sun et al., 2015 and Sanchez-Lorenzo et al., 2017) have been added where the semi-arid problem for CLARA-A1 has been highlighted earlier.

**Repeating specific comment 2:**

**The Section 7 seems to me a bit too quick. The analyses presented in the section (chiefly the one on the change of cloud conditions in the Arctic and Northern Hemisphere) are admittedly conducted as a very preliminary effort. However, even adopting this perspective, the material and figures are integral parts of the paper and would require more studies in order to be published. I am not in favour of including this part. Perhaps the space saved could be effectively allocated to address the previous point of this review.**

**Reply:**

The recommendation from the reviewer to not present this part is obviously strong and should be obeyed.

At the same time, we added this section because we wanted to point out with some examples a couple of application areas which we found especially interesting.  Also, we wanted the manuscript to include a little bit more of scientific discussion to justify it better in the ACP context (we have heard the argument that there are other journals more suitable for sheer descriptions of new data records). Thus, before taking the step to remove the section we want to have the opportunity to provide some more arguments (including the previously

mentioned). After having provided that, we will wait for a final recommendation from the reviewer.

The Arctic surface albedo plots were kind of natural to include with respect to the ongoing concern about the observed reduction of summertime Arctic ice coverage. The new thing here was that it would in our opinion be interesting to also relate it to changes in Arctic cloudiness. Both parameters are available in CLARA-A2 and we just wanted to point that out. The cloud results are certainly not as conclusive as the surface albedo plots but this is nevertheless an interesting piece of information pointing out that ice melting processes do not seem to be directly and highly correlated with changes in cloud cover. The results as such cannot be considered as preliminary (the choice of word here is unfortunate) since they have been produced in exactly the same manner as the surface albedo results, i.e., by use of all monthly averages over the studied period. So, these are final CLARA-A2 results and not preliminary ones. However, this is only a first observation based on monthly and yearly averages which could be complemented by more detailed studies also including additional information (e.g circulation patterns, warm/cold advection conditions, cloud types, etc) to enable a deeper analysis.

Regarding the other example (about global 25-year average trends of cloudiness) we have similar arguments. We just repeated the study (or parts of the study) in the Norris et al. 2016 paper to see what happens if we add another 7 years to the 27 years that they studied. There are obviously differences but also similarities so it is clear that it is difficult to find very clear conclusions. However, the indication that the cloud changes seen by Norris et al., 2016 for mid- and high-latitudes (and which were specifically high-lighted in the conclusions) are maybe not as clear in our study despite having several additional years which would reasonably give better prospects of finding a long-term trend. This would actually call for further and deepened studies here. For this reason we still think this addition to the paper is interesting.

We'll await a final recommendation on the potential removal of Section 7. If that recommendation stands firm we hope that we can at least use the figures (both or one of them) to further illustrate the product groups of clouds and surface albedo in sections 4 and 5.

**Final modification of manuscript:**

We decided to keep Section 7 in the revised manuscript, even if this leads to a significant extension in the length of the paper (considering all other changes). The main reason is that we are keen on seeing more work being done on the exemplified applications so we really want to encourage this.

We removed the statement that results are preliminary and we instead underlined that the provided results are results directly derived from CLARA-A2 data record (**page 14, lines 3-7).** We also emphasized again (**page 15, lines 4-8**) the importance and significance of having a longer time series of data (7 additional years) when repeating the study by Norris et al., (2016). However, we also repeated in both example cases that further in depth studies are needed here to evaluate the full impact of the results (**page 14, lines 29-31**).

Finally, we have also added the correct reference to the paper by Norris et al., (2016) which unfortunately was missing in the first version of the manuscript.

**Repeating specific comment 3:**

**The black/blue colour choice of the dots in the figure (Figure 12) makes them difficult to spot.**

**Maybe a black/red choice would do the job much better.**

**Reply:**

Figure 12 will be revised to increase readability.

**Final modification of manuscript:**

The figure has been updated (new name is Figure 13). Dots have been made bigger and the blue colour has been changed to red.

---

## Author Response (AR2)

Final reply to Co-Editor's recommendation for the ACPD paper

**"CLARA-A2: The second edition of the CM SAF cloud and radiation data record from 34 years of global AVHRR data"**
**by**
**Karl-Göran Karlsson et al.**

**Repeating Recommendation:**

**In conclusion, please go ahead and remove section 7 of the paper. You should put the results and Figures 15 and 16 into section 5 which deals with the CLARA-2 surface albedo product. When transferring this part, please take care and avoid making speculative statements, just comment on the results presented in the paper and discuss them also making reference, whenever appropriate, to others' findings.**

**Another minor point from my side is the clarification of what you refer to as "dynamic aerosol". Please clarify what this "dynamic aerosol" refer to and specify to what exactly it differs with respect to CLARA-1.**

**Reply:**

We have now followed this recommendation. In practice, it means the following:

1. Section 7 has been removed.
2. Reference to Section 7 has been removed in Section 1 (Introduction, lines 28-29 on Page 2 in track change version of manuscript).
3. The mentioning of the study of global redistribution of cloudiness in the abstract has been removed (abstract, line 25 in track change version of manuscript).
4. Reference to Rossow and Schiffer, 1999 is removed since it was only mentioned in the part discussing changes of global cloudiness in Section 7 (page 23, lines 28-29 in track change version of manuscript).
5. The example of decadal changes of surface albedo and cloudiness in the Arctic region has been moved to the end of section 5 (page 12, lines 36-38 and page 13 lines 1-22 in in track change version of manuscript). The associated figures are now numbered 13 and 14.

6. A reference to Norris et al. (2016) is still included (page 8 lines 6-14 in track change version of manuscript) but only for encouraging further studies in this field using CLARA-A2 data.

In addition to this we have also made the following changes:

1. The text on the development of a CLARA-A2 COSP simulator (which was mentioned at the end of the previous section 7) has been moved to the concluding section (page 17 lines 1-3 in track change version of manuscript).
2. A further clarification of the meaning of the introduction of a "dynamic aerosol" has been added (page 11 lines 7-11 in track change version of manuscript).
3. Acronyms IPCC and AR5 have been added to Acronym list (page 18 and 19 in track change version of manuscript).
4. The reference to Jäskeläinen et al (2017) has been modified (page 22 lines 9-12 in track change version of manuscript).
5. Two erroneous entries in Table 3 has been updated/corrected (page 28 in track change version of manuscript).
6. Previous figures 13 and 14 have been renumbered to 15 and 16. All affected references in the text have been updated.

**Final remark:**

**We fully understand the recommendation to remove the discussion of changes in global cloud patterns because of the fact that this is subject to intense scientific discussion (even if the latter originally made us keen on adding something on this to the paper). We hope that you can accept that we keep a reference to the study by Norris et al (2016) in our paper but stay here with just a recommendation to use CLARA-A2 data in future similar studies. The critical comments from the reviewers on this part were also quite justified in the sense that no further attempts to correct the data record for certain artefacts (e.g. due to orbital drift) had been made as opposed to the data being used by Norris et al. (2016). So it is clear that more work is needed here for allowing a direct comparison with previous results in this field.**

**We got the recommendation to 'avoid making speculative statements' in the text we are still using coming originally from the previous Section 7 regarding the Arctic surface albedo and cloud products. In our opinion the statements made here at the end of Section 5 are not speculative but just**

referring to some other findings lately (e.g. by Devasthale et al., 2016). We hope this can be accepted.

[revised manuscript text omitted]